

# Basal sauropodomorph locomotion: ichnological lessons from the Late Triassic trackways of bipeds and quadrupeds (Elliot Formation, main Karoo Basin)

Lara Sciscio[1,2,3], Emese M. Bordy[2], Martin G. Lockley[4] and Miengah Abrahams[2]

[1] Department of Geoscience, University of Fribourg, Fribourg, Switzerland
[2] Department of Geological Sciences, University of Cape Town, Cape Town, South Africa
[3] JURASSICA Museum, Porrentruy, Jura, Switzerland
[4] Dinosaur Trackers Research Group, University of Colorado, Denver, Colorado, United States of America

Corresponding author
Lara Sciscio, l.sciscio@gmail.com

## ABSTRACT

Using modern ichnological and stratigraphic tools, we reinvestigate two iconic sauropodomorph-attributed tetradactyl ichnogenera, *Pseudotetrasauropus* and *Tetrasauropus*, and their stratigraphic occurrences in the middle Upper Triassic of Lesotho. These tracks have been reaffirmed and are stratigraphically well-constrained to the lower Elliot Formation (Stormberg Group, Karoo Basin) with a maximum depositional age range of <219–209 Ma (Norian). This represents the earliest record of basal sauropodomorph trackways in Gondwana, if not globally. Track and trackway morphology, the sedimentary context of the tracks, and unique features (*e.g.*, drag traces) have enabled us to discuss the likely limb postures and gaits of the trackmakers. *Pseudotetrasauropus* has bipedal (*P. bipedoida*) and quadrupedal (*P. jaquesi*) trackway states, with the oldest quadrupedal *Pseudotetrasauropus* track and trackway parameters suggestive of a columnar, graviportal limb posture in the trackmaker. Moreover, an irregularity in the intermanus distance and manus orientation and morphology, in combination with drag traces, is indicative of a non-uniform locomotory suite or facultative quadrupedality. Contrastingly, *Tetrasauropus*, the youngest trackway, has distinctive medially deflected, robust pedal and manual claw traces and a wide and uniform intermanus distance relative to the interpedal. These traits suggest a quadrupedal trackmaker with clawed and fleshy feet and forelimbs held in a wide, flexed posture. Altogether, these trackways pinpoint the start of the southern African ichnological record of basal sauropodomorphs with bipedal and quadrupedal locomotory habits to, at least, c. 215 Ma in the middle Late Triassic.

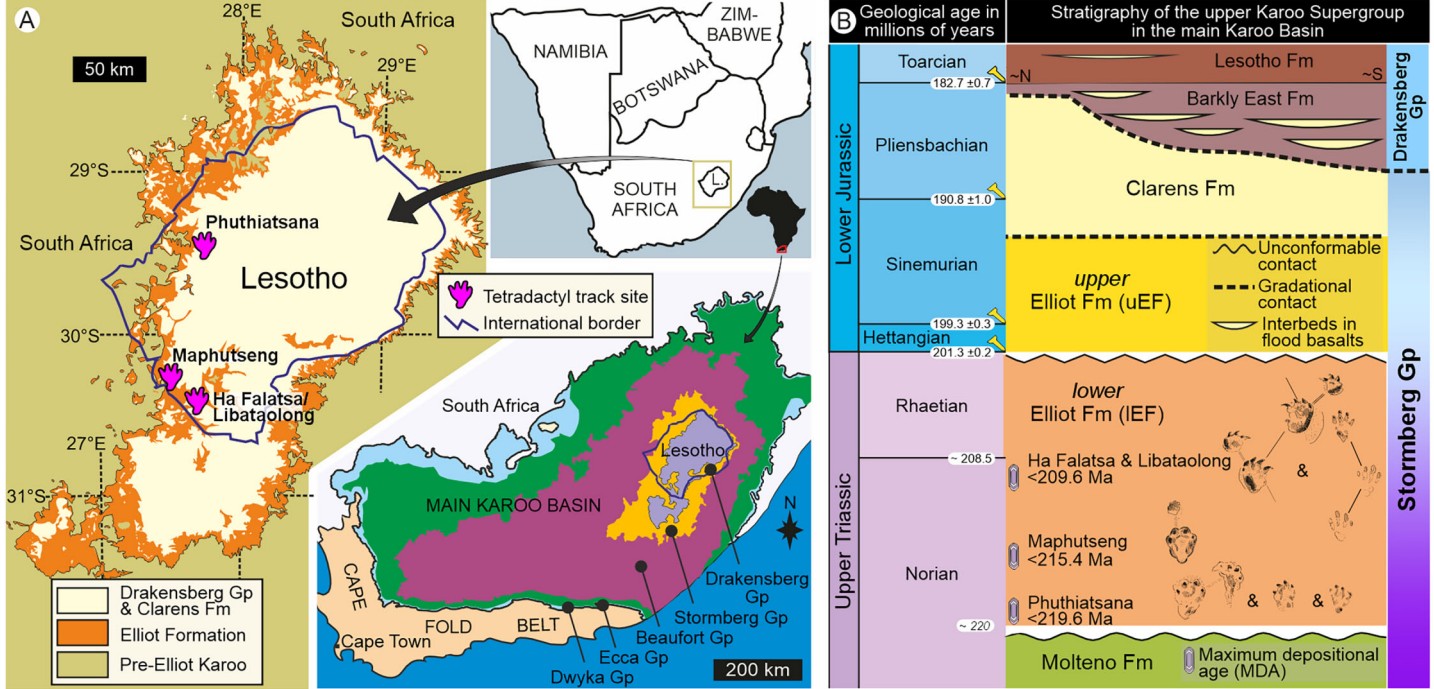

**Figure 1 Geological context of the studied large tetradactyl track sites in Lesotho.** (A) Location of the four track sites within the geological map of the upper Karoo Supergroup and the main Karoo Basin. (B) Stratigraphical context, including main track types and maximum depositional ages of the Late Triassic sites within the lower Elliot Formation and the Stormberg Group. Abbreviations: Gp, Group; Fm, Formation.

## INTRODUCTION

Basal sauropodomorphs (non-sauropodan sauropodomorphs; also referred to as "prosauropods") represent a group of terrestrial herbivorous dinosaurs that were common in southern Gondwana in the Late Triassic and Early Jurassic, at the dawn of dinosaurian radiation. Basal sauropodomorphs are considered the forerunners of sauropods, the diverse and ubiquitous group of large, habitually quadrupedal non-avian dinosaurs that dominated the rest of the Mesozoic (*McPhee et al., 2020*).

Globally, and in southern Africa (Fig. 1), Late Triassic bipedal and quadrupedal basal sauropodomorphs are known from the body fossil record (*e.g., McPhee et al., 2017, 2020; Pol et al., 2021; Apaldetti et al., 2021*) and the ichnological record (*e.g., Ellenberger & Ellenberger, 1956a; Ellenberger, 1972; Lallensack et al., 2017; Klein & Lucas, 2021; Lockley et al., 2023*). While basal sauropodomorphs are, generally, considered smaller-bodied and defined by their obligatory bipedalism, facultative to habitual quadrupedality appears to have evolved by the Late Triassic (*e.g., Lallensack et al., 2017; Apaldetti et al., 2018*). This locomotory adaptation is often linked to increases in body size and coupled with forelimb robusticity in the absence of columnar limb posture (*e.g., Lessemsaurus, Ledumahadi,* and *Ingentia; Apaldetti et al., 2018; McPhee et al., 2017, 2018; Pol et al., 2021*). The latter, along with unflexed limbs and other adaptations, enabled sauropod dinosaurs to increase body size and evolve graviportalism without compromising their physiology (*e.g., Sander & Lallensack, 2018*). Understanding the locomotion of ancestral forms of sauropods is

important for unlocking how sauropods developed their specialized form of quadrupedality, which had both palaeobiological and palaeoecological consequences for post-Triassic sauropod dominance. The southern African dinosaurian ichnological record can shed light on this pivotal transition within sauropodomorph evolution.

Our work represents a thorough re-examination of several iconic basal sauropodomorph trackways related to the ichnogenera *Pseudotetrasauropus* and *Tetrasauropus* within the Upper Triassic (Norian) Elliot Formation of Lesotho (Fig. 1). Previously, these tetradactyl trackways were only known through brief reports published early in the latter half of the twentieth century (*e.g.*, *Ellenberger & Ellenberger, 1956a*, *1956b*, *1958*, *1960*; *Ellenberger et al., 1963*; *Ellenberger, Ellenberger & Ginsburg, 1970*; *Ellenberger, 1970*, *1972*) and more recently, their casts made during that time (*D'Orazi Porchetti & Nicosia, 2007*). Our reassessment of these trackways and tracksites also includes the re-examination of the stratigraphy, geochronology, and sedimentology of the host sedimentary rocks. The aims of this article are to provide a holistic appraisal of the trackways and track-bearing sites, to establish if these trackways belong to a single ichnotaxon or represent different ichnotaxa and evaluate the different locomotor habits adopted by the trackmakers. This work has significant implications for the widespread distribution, ichnotaxonomy, and inferred affiliations and locomotor styles of trackways attributed to basal sauropodomorphs.

## Palaeontological and ichnological background

Basal sauropodomorphs are represented by an extensive southern African body fossil record from the Upper Triassic lower Elliot Formation (lEF; Fig. 1) and its equivalents. There are eight recognized lEF taxa with both assumed quadrupedal and bipedal postures: *Blikanasaurus cromptoni*, *Melanorosaurus readi*, *Plateosauravus cullingsworthi*, *Eucnemesaurus entaxonis*, *Eucnemesaurus fortis*, *Meroktenos thabanensis*, *Sefapanosaurus zastronensis*, and *Kholumolumo ellenbergerorum* (*e.g.*, *McPhee et al., 2017*; *Peyre de Fabrègues & Allain, 2016*, *2019*; *Bordy et al., 2020*; *Viglietti et al., 2020*). Additionally, the lEF has a relatively well-documented tetradactyl trace fossil record with several forms attributed to basal sauropodomorphs (*e.g.*, *Ellenberger & Ellenberger, 1956a*, *1958*, *1960*; *Ellenberger et al., 1963*; *Ellenberger, Ellenberger & Ginsburg, 1970*; *Ellenberger, 1970*, *1972*; *Viglietti et al., 2020*). Together, these fossil records serve as archives for hind- and forelimb anatomy, pedal postures, and increasing body size in the forerunners of sauropods (*e.g.*, *Bonnan & Yates, 2007*; *Yates et al., 2010*; *McPhee et al., 2018*; *Apaldetti et al., 2021*).

Late Triassic–Early Jurassic tetradactyl tracks attributed to basal sauropodomorphs have a Pangaean distribution (*e.g.*, *Lockley, Lucas & Hunt, 2000*; *Lockley et al., 2001*, *2023*; *Marsicano & Barredo, 2004*; *Masrour & Pérez Lorente, 2014*; *Niedźwiedzki, 2011*; *Niedźwiedzki et al., 2017*; *Lallensack et al., 2017*; *Xing et al., 2018*; *Klein & Lucas, 2021*; *Mukaddam et al., 2021*; *Falkingham et al., 2022*) and are mostly assigned to the ichnotaxa *Otozoum*, *Pseudotetrasauropus*, *Evazoum*, *Kalosauropus* (the OPEK plexus within the ichnofamily Otozoidae; *Lockley, Lucas & Hunt, 2006a*, *2006b*; *Lockley et al., 2023*), *Tetrasauropus*, and *Eosauropus* (overview in *Klein & Lucas, 2021*). In southern Africa (Fig. 1), Upper Triassic large tetradactyl tracks are only known from the lEF and were first

documented by *Dornan (1908*; Morija, Lesotho) with their systematic recording begun in the 1950s by the legendary tracker, Paul Ellenberger (*e.g.*, *Ellenberger & Ellenberger, 1956a*, *1958*, *1960*; *Ellenberger et al., 1963*; *Ellenberger, Ellenberger & Ginsburg, 1970*; *Ellenberger, 1970*, *1972*). Of interest herein are the ichnogenera *Pseudotetrasauropus* and *Tetrasauropus* (*Ellenberger, 1972*), with their stratigraphic and sedimentological context briefly described by *Ellenberger (1970*, *1972)*, who placed them in the upper Molteno Formation ("Zones A3 and A4").

*Pseudotetrasauropus* was described by *Ellenberger (1972*, p. 40) as having a tetradactyl pes with an open fan shape, digit I laterally mobile relative to the other digits that are connected until the penultimate phalanx and by the pedal sole. The digits end in blunt claw traces and *Ellenberger (1972)* provides a phalangeal formula (1)-2-3-3-1. *Pseudotetrasauropus* has a total of eight ichnospecies from four locations: *P. acutunguis* (Phuthiatsana), *P. augustus* (Phuthiatsana), *P. jaquesi* (Phuthiatsana), *P. bipedoida* (Libataolong), *P. dulcis* (Seaka/Ha Falatsa), *P. elegans* (Seaka/Ha Falatsa), *P. francisci* (Seaka/Ha Falatsa), *P. mekalingensis* (Mekaling). The initial attempt to stabilize this knotty ichnotaxonomy, using published descriptions, was undertaken by *Olsen & Galton (1984)* who reduced the ichnodiversity by synonymizing *Pseudotetrasauropus* with *Brachychirotherium* and retained *Tetrasauropus* as a valid ichnogenus. Later, *Lockley & Meyer (2000)* stated that *Pseudotetrasauropus* "is essentially identical to" *Otozoum*, a similarity recognized by Ellenberger (*Ellenberger & Ellenberger, 1958*, p. 67; *Ellenberger et al., 1963*, p. 317) for *P. bipedoida* at Libataolong (*Ellenberger, 1972*). *Lockley & Meyer (2000)* considered six *Pseudotetrasauropus* ichnospecies to represent bipeds with *P. elegans* and *P. jaquesi* representing quadrupeds. *Rainforth (2003)* and *D'Orazi Porchetti & Nicosia (2007)* viewed *Pseudotetrasauropus* and *Otozoum* as being distinctly different ichnotaxa. This assertion is based on the more discrete, separated digits of *Pseudotetrasauropus* along with its lack of a metatarsal-phalangeal pad, which is a distinctive feature of the complex pads of *Otozoum* (see *Rainforth, 2003*, figs. 3C, 3D; *Mukaddam et al., 2021*, figs. 6, 7).

The recent balanced and systematic re-investigation of these tetradactyl ichnotaxa, using their casts in the Ellenberger Collection (University of Montpellier, France), lead to a conservative treatment of *Pseudotetrasauropus* (*D'Orazi Porchetti & Nicosia, 2007*). This resulted in *P. bipedoida* as only valid ichnogenus and species, with *P. augustus* its synonym, *P. jaquesi* referable to ?*Lavinipes*, and *P. curtus*, *P. francisci*, *P. acutunguis* and *P. elegans* as *nomina dubia*. However, the Phuthiatsana *Pseudotetrasauropus* casts could not be located and were not physically assessed by *D'Orazi Porchetti & Nicosia (2007)*. *Tetrasauropus* cast material from Ha Falatsa also lead *D'Orazi Porchetti & Nicosia (2007)* to validate this ichnogenus, diagnosed by its entaxonic footprint and prominent, medially oriented claw traces in both the pes and manus tracks.

## MATERIALS AND METHODS

Sedimentology and track data were collected in the field (Figs. 1, 2) under the field permits (NR/M/E/10 and MTEC7/33) issued by the Lesotho Government Department of Mines and Geology and Ministry of Tourism, Environment and Culture. Other track data utilized the original plaster of Paris cast replica collections housed at the Morija Museum and

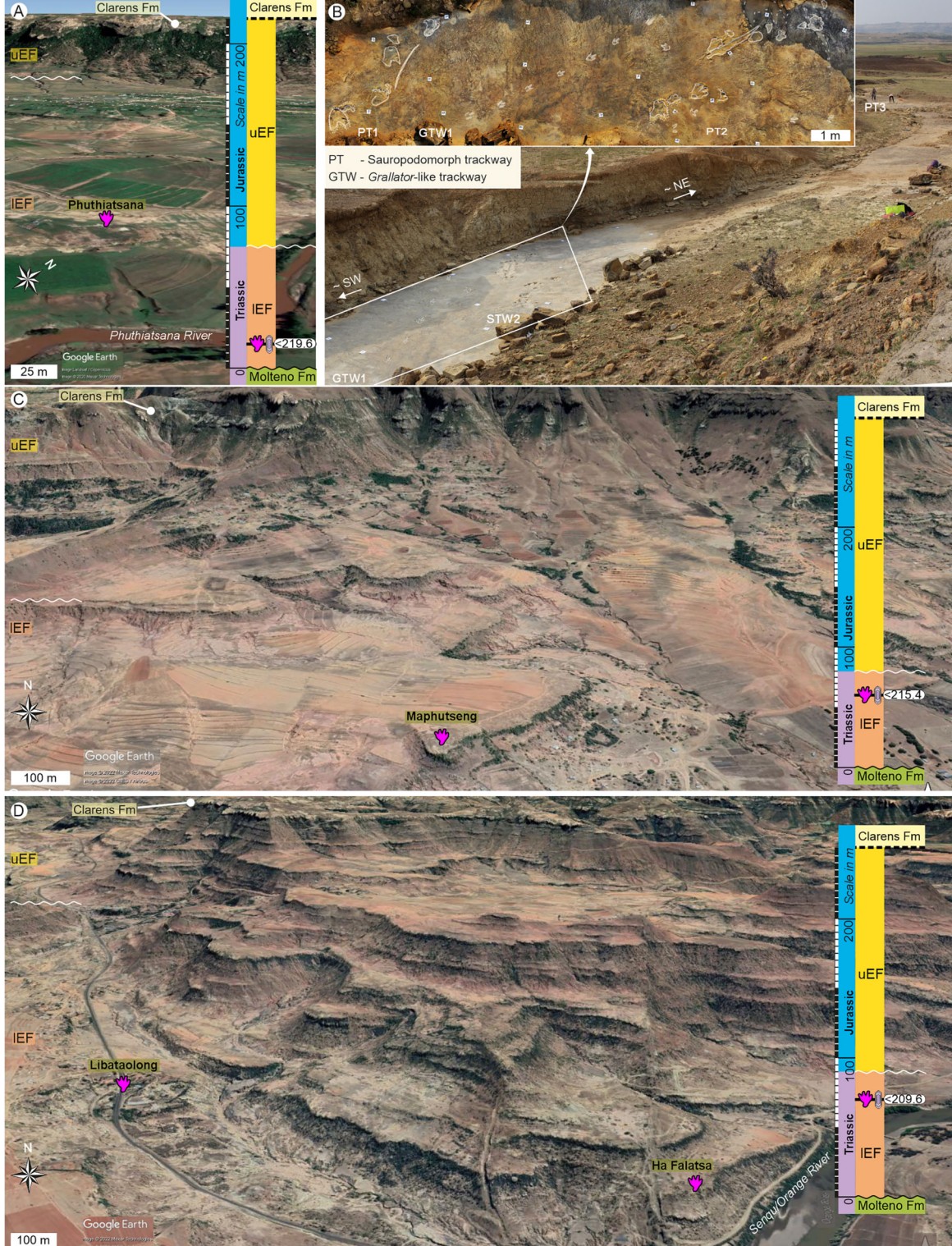

**Figure 2 Field views of the large tetradactyl track sites showing their stratigraphic context and the position of the track-bearing surfaces within the lower Elliot Fm.** (A) Phuthiatsana track site. (B) Close-up field views of the Phuthiatsana track site. (C) Maphutseng track site. (D) Ha Falatsa–Libataolong track sites that are <1.7 km apart and in identical stratigraphic level within the lEF. For legend and abbreviations, see Fig. 1. Map data © 2023 Google.

**Table 1 Norian lower Elliot Formation tetradactyl trackways measurements.** Trackways are from Phuthiatsana (PT1, PT2, PT3), Maphutseng (MP1), Ha Falatsa (HF1) and Libataolong (LB1).

| | PL (cm) | PW (cm) | PL/PW | I^IV (°) | ML (cm) | MW (cm) | ML/MW | Pes Pace (cm) | Stride λ (cm) | Pes angulation (°) | Manus Pace (cm) | Manus angulation (°) | GAD | h (m) | Pes gait (λ/h) | Trackway width | Speed, (m.s⁻¹) | Speed (km/h) | HI |
|---|---|---|---|---|---|---|---|---|---|---|---|---|---|---|---|---|---|---|---|
| PT1 ($n_p$ = 4; $n_m$ = 4) | 56 | 56 | 1.0 | 61 | 12 | 29 | 0.4 | 118 | 216 | 138 | 118 | 32 | 1.2 | 2.2 | 1.0 | 82 | 1.1 | 4.0 | 11 |
| PT2 ($n_p$ = 5; $n_m$ = 5) | 50 | 42 | 1.2 | 63 | 14 | 29 | 0.5 | 130 | 248 | 133 | 131 | 97 | 1.4 | 2.0 | 1.2 | 85 | 1.6 | 5.7 | 19 |
| PT3 ($n_p$ = 7; $n_m$ = 0) | 46 | 36 | 1.3 | 51 | | | | 116 | 164 | 131 | | | | 1.8 | 0.9 | 90 | 0.9 | 3.1 | |
| MP1 ($n_p$ = 4; $n_m$ = 4) | 45 | 43 | 1.1 | 55 | 13 | 28 | 0.5 | 127 | 190 | 99 | 118 | 113 | 1.4 | 1.8 | 1.0 | 104 | 1.1 | 4.1 | 18 |
| HF1 ($n_p$ = 5; $n_m$ = 5) | 45 | 41 | 1.1 | 56 | 30 | 21 | 1.4 | 112 | 188 | 116 | 153 | 81 | 1.3 | 1.8 | 1.1 | 92 | 1.2 | 4.3 | 34 |
| LB1 ($n_p$ = 3; $n_m$ = 0) | 40 | 33 | 1.2 | 57 | | | | 71 | 116 | 109 | | | | 1.6 | 0.7 | 61 | 0.6 | 2.1 | |
| Average | 47 | 42 | 1.1 | 57 | 17 | 27 | 0.7 | 112 | 187 | 121 | 130 | 81 | 1.3 | 1.9 | 1 | 86 | 1.08 | 3.9 | 21 |

**Note:**

Abbreviations as stated in text; $n_p$/$n_m$ = number of pes/manus, respectively.

Archives (M&A; Lesotho; labelled BM) and in the Ellenberger Collection at the University of Montpellier (UoM; France; labelled UM LES or LES). Silicon replicas made of two tracks (specimen numbers: BP/6/736—Phuthiatsana "*P. jaquesi*"; BP/6/742—Ha Falatsa "*T. unguiferus*") are housed in the Ichnology Collection of the Evolutionary Studies Institute (ESI) at the University of Witwatersrand, South Africa. Several original Ellenberger documents (field notes, maps, photographs, specimen descriptions for display cabinets, *etc.*), archived but uncatalogued at Morija M&M and UoM, were also investigated. Morphological preservation and associated grading (MP scale; *Belvedere & Farlow, 2016*; *Marchetti et al., 2019*) was used as a reference for anatomical fidelity (*Gatesy & Falkingham, 2017*; Table S1).

Track measurements (Table 1) were taken in the field for all sites except Libataolong (measured casts using ImageJ v1.53t or reference photographs) and include pes/manus length (PL/ML), pes/manus width (PW/MW), length of digits (I, II, III, IV), interdigital angles (I^IV), pace length, stride length (SL), and pes/manus pace angulation (P-/M-ANG). The gleno-acetabular distance (GAD) was measured from trackways as shown in *Marty (2008)*. Pes length is considered as the length from the tip of digit III to the base of the proximal phalangeal pad ("heel"), and PW is from the outer margin of digit IV to digit I. The pes pace angulation is taken from digit III. Trackway width (TW) was measured from the external margins of opposite and consecutive steps. Photogrammetric 3D models were made for trackway surfaces at each site following steps detailed in *Mallison & Wings (2014)*. AgisoftPhotoscan (standard version 1.1.4) software was used to process point clouds and 3D models were converted to elevation maps in the open-source CloudCompare (v.2.6.1) and ParaView (v.5.10.0). All digital models are deposited at MorphoSource in accordance with *Falkingham et al. (2018)*. The 2D representations of selected holotype trackways were collected (tracings on transparent film) and are archived in the University of Colorado Museum collections (tracings T1899–T1902, T1904, and T1906).

Estimations of the hip height (*h*) used average pes length (PL) by assuming that *h* is c. four times pes length (*Alexander, 1976*). Gait (λ) was estimated using average pedal stride lengths and calculated hip heights where λ/*h* < 2.0 indicates walking (*Thulborn & Wade, 1984*). The estimated walking speed could be determined using *Alexander (1976)*:
$u = 0.25 \, g^{0.5} \times \lambda^{1.67} \times h^{-1.17}$; where, g = gravitational acceleration in m/sec, $\lambda$ = stride length, and $h$ = hip height. The heteropody index (HI) of *Gonzalez Riga & Calvo (2009*; HI = [(ML × MW)/(PL × PW] × 100) and pes trackway ratio (PTR) of *Romano, Whyte & Jackson (2007)* were used as size gauging proxies (Tables 1 and S1).

## RESULTS

### Stratigraphy of the Norian tetradactyl tracks

Upon relocating these unique tetradactyl track sites (Phuthiatsana, Maphutseng and Ha Falatsa-Libataolong) in Lesotho, our detailed geological mapping, in combination with previous work undertaken (sedimentology, magnetostratigraphic investigations and geochronological dating) at the track-bearing surfaces, revealed that they differ in age and are part of the Norian section of the lEF, spanning a time interval of c. 10 Ma (Figs. 1 and 2; *Bordy, Hancox & Rubidge, 2004a*, *2004b*; *Bordy et al., 2020*).

The Phuthiatsana track-bearing surface is >80 m long and is located <1 km to the southwest of the Phuthiatsana River bridge on road B21 (GPS: 29°21′30.02″S, 27°36′35.61″E; Figs. 1, 2A and 2B). Radioisotopic dating of detrital zircons from the site yielded a maximum depositional age of 219.6 ± 2.5 Ma, indicating that the host rocks formed in the middle Norian (Figs. 1B, 2A and 2B; *Bordy et al., 2020*). Stratigraphically, this is the oldest tracksite among the four sites described herein and is in the lowermost Elliot Formation (Figs. 1B, 2).

The Maphutseng track-bearing surface is c. 80 m long and is located c. 1 km to the west of the Bethesda Mission in the Maphutseng River valley (GPS: 30°12′46.66″S 27°28′51.45″E; Figs. 1, 2A and 2B). Radioisotopic dating of detrital zircons from the site yielded a maximum depositional age of 215.4 ± 2.5 Ma, indicating that the host rocks formed in the middle-late Norian (Figs. 1B, 2A and 2B; *Bordy et al., 2020*). Stratigraphically, this is the second oldest tracksite among the four sites. Located in the lEF, the site is c. 15 m below the contact with the upper Elliot Formation (Figs. 1B, 2C) and c. 20 m above the Maphutseng bone bed. The latter was discovered in 1955 and yielded several hundred fossils of mostly basal sauropodomorphs (*i.e.*, *Kholumolumo ellenbergerorum*, *Peyre de Fabrègues & Allain, 2019*) but also cynodonts and amphibians.

The Ha Falatsa-Libataolong tracksite comprises two sites, found <1.7 km apart at Ha Falatsa and Libataolong (Figs. 1, 2A and 2C). The Ha Falatsa track-bearing surface, herein labelled HF1, is a few metres long and is located within a larger track site that extends over 850 m from the old Senqu (Orange) River bridge (aka 'Seaka Bridge') to just beyond HF1 (GPS: 30°22′11.02″S, 27°34′1.87″E). The larger track site contains multiple tridactyl, tetradactyl and pentadactyl ichnotaxa on several co-genetic track-bearing surfaces that outcrop along the west bank of the Orange River (Figs. 1B, 2D) atop the same lEF sandstone unit. These ichnotaxa are provenanced as Seaka (shorthand for 'Seaka Bridge') and Ha Falatsa in *Ellenberger (1970*, *1972)*. The Ha Falatsa site yielded a maximum depositional age of 209.6 ± 1.4 Ma, indicating that the host rocks formed in the late Norian (Figs. 1B, 2C; *Bordy et al., 2020*). The Libataolong track-bearing surface with its two tetradactyl trackways, herein labelled LB1 and LB2, was discovered in the early 1950s (*Ellenberger & Ellenberger, 1956a*, *1958*), but has been subsequently partially destroyed
during the construction of the A2 main road. However, the site with its few remaining tridactyl tracks was relocated and mapped by us to be in the same stratigraphic level as HF1 at Ha Falatsa (GPS: 30°21′52.14″S 27°33′0.37″E; Figs. 1B, 2D). Stratigraphically, the Ha Falatsa-Libataolong track site is the youngest among the four tracksites described herein (Figs. 1B, 2D).

## PHUTHIATSANA TETRADACTYL TRACKS (<219.6 MA)

At Phuthiatsana, three tetradactyl trackways were studied. Two trackways (PT1 and PT2; Fig. 2B; Table 1) are located at the southwestern end of palaeosurface and are semi-parallel to one another. The third trackway (PT3; Fig. 2B; Table 1) lies c. 20 m to the northeast.

### Trackway PT1

PT1 comprises four tetradactyl pes impressions that are asymmetrical, large (av. PL of 56 cm, av. PW of 56 cm, av. PL/PW of 1.0, av. surface area of c. 1,068 cm$^2$; Fig. 3, Tables 1 and S1), and digitigrade to semi-plantigrade (*i.e.*, pes preserves the last digital phalange and claw traces of II–IV). The digits, which have a subrounded distal digit morphologies ending in blunt claw impressions and wide total digit divarication angles of 62°, are distinct from the slightly raised sole impressions (Figs. 3–5). The shorter digit I is medially directed and well separated from digits II–IV. The latter are more anteriorly directed, with digit III the longest, and digits II–IV having subequal lengths (I < II = IV < III). Digit I impressions potentially bear a phalangeal and metatarsaophalangeal pad trace (*e.g.*, Fig. 4B) and a sickle-shaped claw trace suggesting the claw was medially projected and held parallel to the substrate (Fig. 4). In track LP1, this claw trace is exaggerated due to the digit movement in a saturated substrate during the pushoff phase. The total sole impression is elongated to lunate in shape giving the impression of a medial longitudinal arch. The rear of the sole preserves a deeper, oval "heel" mark that is longer than wide, with an av. distance of 27.3 cm from its anterior margin to the tip of digit IV. This could either represent a large digit V pad or the digital metatarsal region. Metatarsal-phalangeal pads are otherwise absent with only a longitudinal trace marking the external impression of digit IV.

PT1's tetradactyl manus (av. MW 29.3 cm; av. ML 12 cm) are wider than long (av. ML/MW = 0.4, with surface area of c. 129 cm$^2$), very moderately supinated with rounded margins and clawless digit impressions (Figs. 3, 4). Digit impressions are arranged in an open arc shape accentuated by the concave proximal margin. The manus tracks are either anteriorly orientated distally and parallel to the pes or anterolaterally turned outward (Fig. 3). The push-back ridges created by the manus are partially obscured by the expulsion rims of the pes, with the manus-pes pair track (LM3) showing the least overprinting (Figs. 3, 4).

Overall, the PT1 trackway is a straight, 3.8 m long trackway composed of four large, consecutive pes tracks and three manus impressions, showing strong heteropody (Fig. 3). The long axis of the pes, through digit III, is roughly parallel to the trackway midline and the manus is slightly outwardly rotated, lying c. 20–30 cm above the pes. The claw impression on pes digit I is inwardly rotated. The overall width of the trackway is c. 82 cm

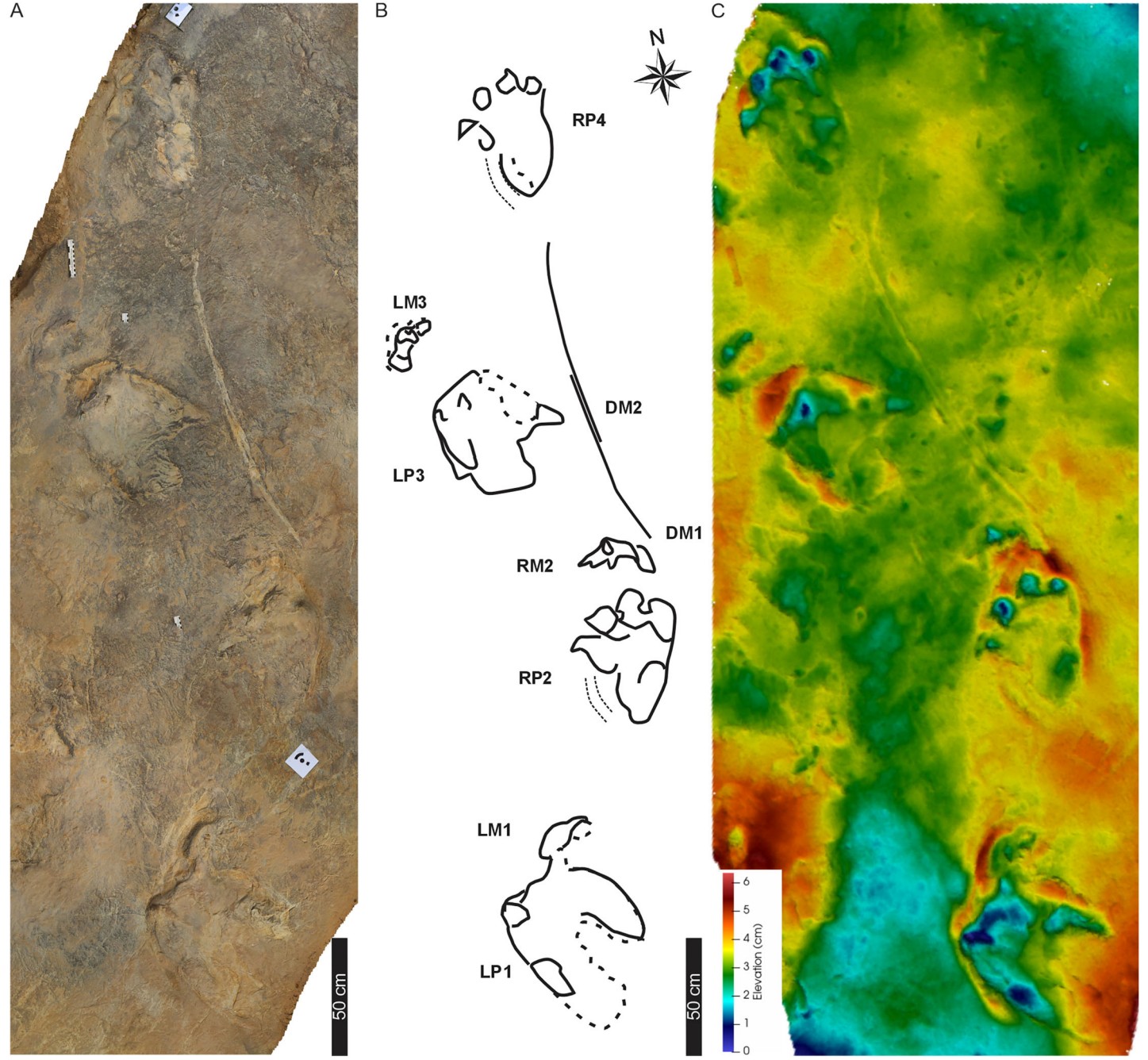

**Figure 3 Trackway PT1 at the Phuthiatsana track site.** (A) Orthophotograph, (B) interpretative outline drawing with labelled pes, manus and drag traces. The arcuate drag traces (DM1, DM2) are between the second and the fourth pes-manus pairs; and (C) false-colour depth map of PT1 (depth increases with cooler colours). Dashed lines indicate soft sediment deformation. For more details, see Fig. 4. For location, see Fig. 2B. Abbreviations as per text.

with PTR of 69% ("narrow gauge"; *Romano, Whyte & Jackson, 2007*). The average pes pace and stride lengths are 118 and 216 cm, respectively (Table 1). Intermanus distance is variable but comparable to the interpes distance (av. manus pace 118 cm). The estimated *h*

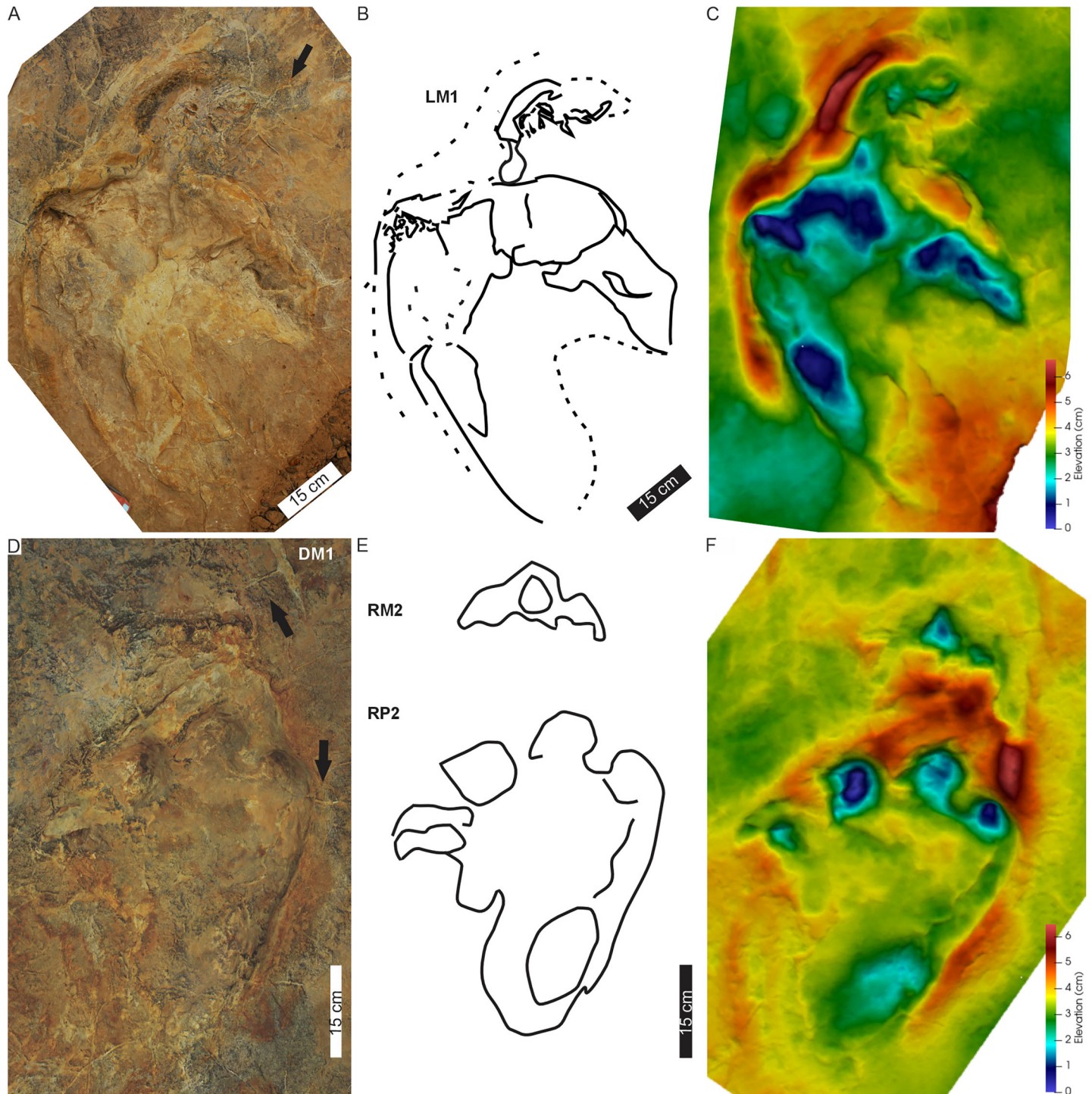

**Figure 4** **Pes-manus pairs in Phuthiatsana trackway PT1.** (A) Orthophotograph, (B) interpretative outline drawing, and (C) false-colour depth map of the first left pes-manus pair. (D) Orthophotograph, (E) interpretative outline, and (F) false-colour depth map of the second right pes-manus pair. Black arrows indicate desiccation cracks. Note similar distribution in colour depth map showing deeply impressed digits of the pes and a raised sediment ridge overprinting the manus impression. Dashed lines indicate soft sediment deformation.

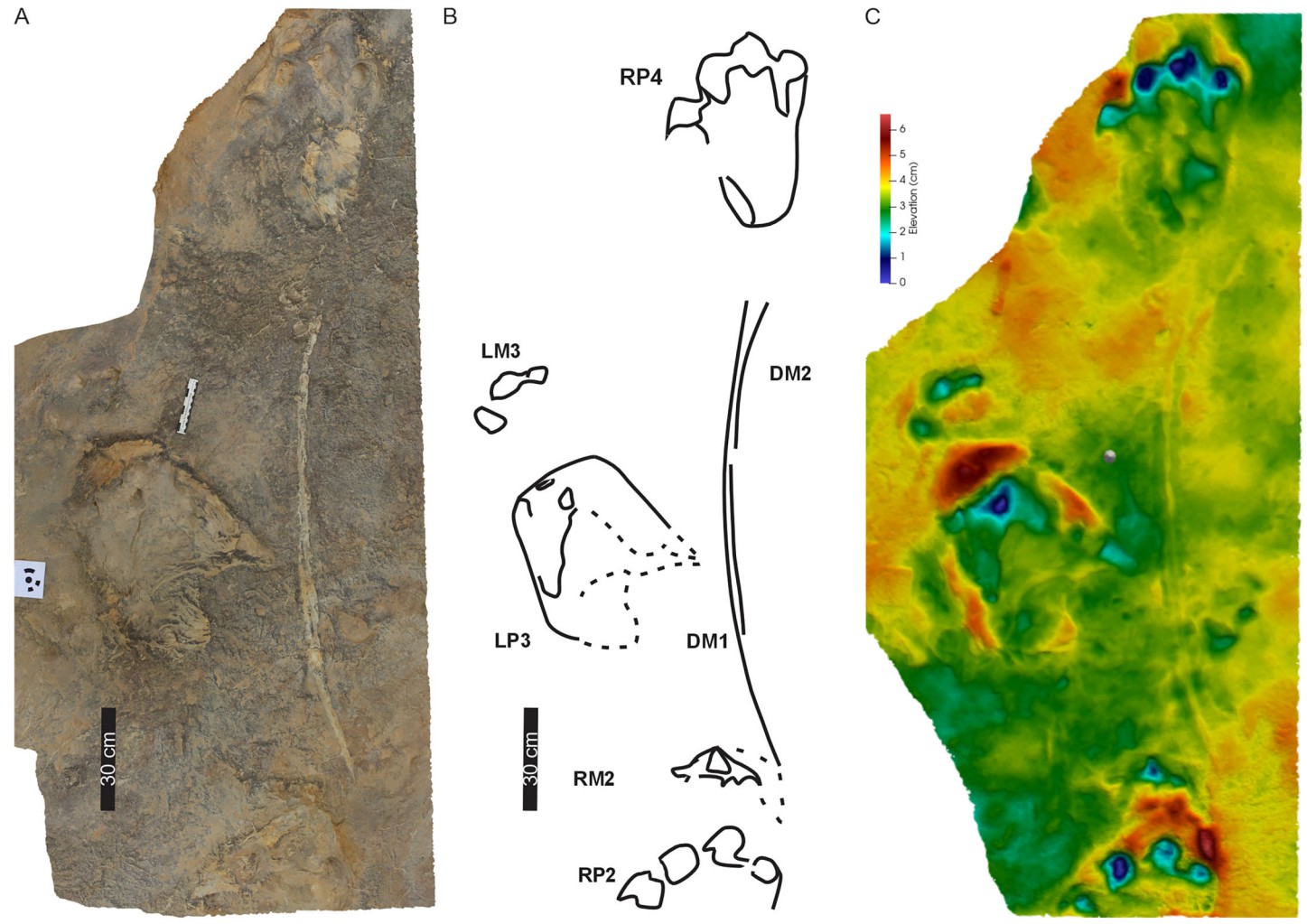

**Figure 5** **Drag traces, DM1 and DM2, associated with Phuthiatsana trackway PT1.** (A) Orthophotograph, (B) labelled line-drawing overlay, and (C) false-colour depth map of drag trace (DM2) between the penultimate manus track (RM2) and the PT1-RP4 pes-manus pair. Dashed lines indicate soft sediment deformation.

of the quadrupedal trackmaker is 2.2 m, GAD is 1.2 m, gait is <2.0 (0.6) with a walking speed of 4 km/h (Table 1).

**Drag traces:** PT1 is associated with two parallel and discontinuous arcuate drag traces (DM1 and DM2; Fig. 5). These curve from the external margin of the second manus impression (Figs. 4B, 5) towards the trackway midline and outwardly towards the base of RP4. DM1 and DM2 are narrow (c. 2–4 cm wide) and short traces (lengths of c. 85 and 91 cm long, respectively). DM1 is U-shaped in cross-section but becomes W-shaped medially (Fig. 5). DM2 starts c. 50 cm from the proximal end of DM1, is a U-shaped trace that arcs towards the outer edge of RP4 (Fig. 5). Expulsion rims are noticeable along the length of both DM1 and DM2.

**Substrate consistency and impression depth:** Soft-sediment deformation and collapse structures modify PT1 tracks and indicate variation in substrate coherence. This is more evident on the left-hand side of the trackway, with penetrative tracks showing plastic

deformation. The latter exaggerates the presence and orientation of a recurved digit I trace in the consecutive left pes tracks (Figs. 3–5). Moreover, the obliquely orientated digit I projected its long axis forward and downward into the substate, creating an 'upfold' of sediment upon digit extraction furthering the appearance of being medially elongate (Fig. 4A). Each pes track, despite substrate pliability, has sediment displacement rims in front of digits II–IV, where the sediment was pushed not only upward into a marginal ridge, but also forward and towards the manus impression (Figs. 3–5). A ridge of sediment is also pushed upwards and away from the external margin of digit IV. A rotational component is noticeable with digit I on the righthand side of the trackway, where a groove (Fig. 3C) shows the arcing curve of digit I and its claw trace. Manus impressions exhibit a posterior, smaller up- and backward-pushed ridge of sediment in contrast to the anterior of the impression, which lacks sediment distortion. The depth of the pes tracks is greatest within the distal digits and less pronounced along the external margin of digit IV.

The pes-sole impression is slightly raised (*e.g.*, RP2, RP4) and the digits' tips are distinct. Depth of the manual impressions is equal to and slightly shallower than those of the pes (Figs. 3–5).

### PT2 trackway

PT2 comprises five large (av. TL of 49.8 cm, av. TW of 42.3 cm, surface area of c. 1,170 cm$^2$), tetradactyl pes impressions that are longer than wide (av. PL/ML of 1.2 ± 0.3) and digitigrade to semi-plantigrade (Figs. 6, 7; Table 1; Figs. S1A–S1H, S2). The pes tracks have a rounded inverted trapezoid shape with deep oval impressions, which define the posterior margin (*i.e.*, "heel"), and are 30.7 cm from the tip of digit IV. The relatively short and subrounded rectangular digits are partially impressed along their length, with the digit tips being defined by rectangular margins indicating blunt claw traces (I < II = IV < III). Digits II–IV have a forward orientation, while digit I is anteriorly directed. The total digit divarication angle average is 63° and no metatarsal-phalangeal pads are preserved.

The tetradactyl, crescent-shaped manus of PT2 (av. MW 29.3 cm; av. ML 14 cm; av. ML/MW = 0.5, with surface area of c. 370 cm$^2$) has rounded external margins. The manus morphology, degree of supination and position along the trackway and relative to the trackway midline is irregular (Figs. 6, 7). The degree of manus rotation, relative to the pes, varies between the consecutive left and right sides of the trackway. For example, PT2-LM1 supination has preserved a partial 'palm' impression separated by a ridge of sediment from the adjacent arc of digits depressions. Additionally, the consecutive right manus tracks are displaced to the right of the corresponding pes; contrastingly, the successive left manus tracks are placed proximal to and above the digit III of the pes. An extramorphological feature (Fig. 7B), on manus impression PT2-RM2 and PT2-RM4, of an oval-shaped depression (c. 8 cm length, 5 cm width) above the general arc of the digits is noted.

Overall, PT2 is a c. 4 m straight to curving trackway showing heteropody (Figs. 5–7; Table 1) with the long axis of the pes tracks (through digit III) parallel to the trackway axis. A defining character of PT2 is the variable and pronounced manus placement and supination. The long axis of the manus can be parallel, rotated inwards by c. 60° or outwards by c. 40° relative to the trackway axis. The stride and pace lengths vary based on
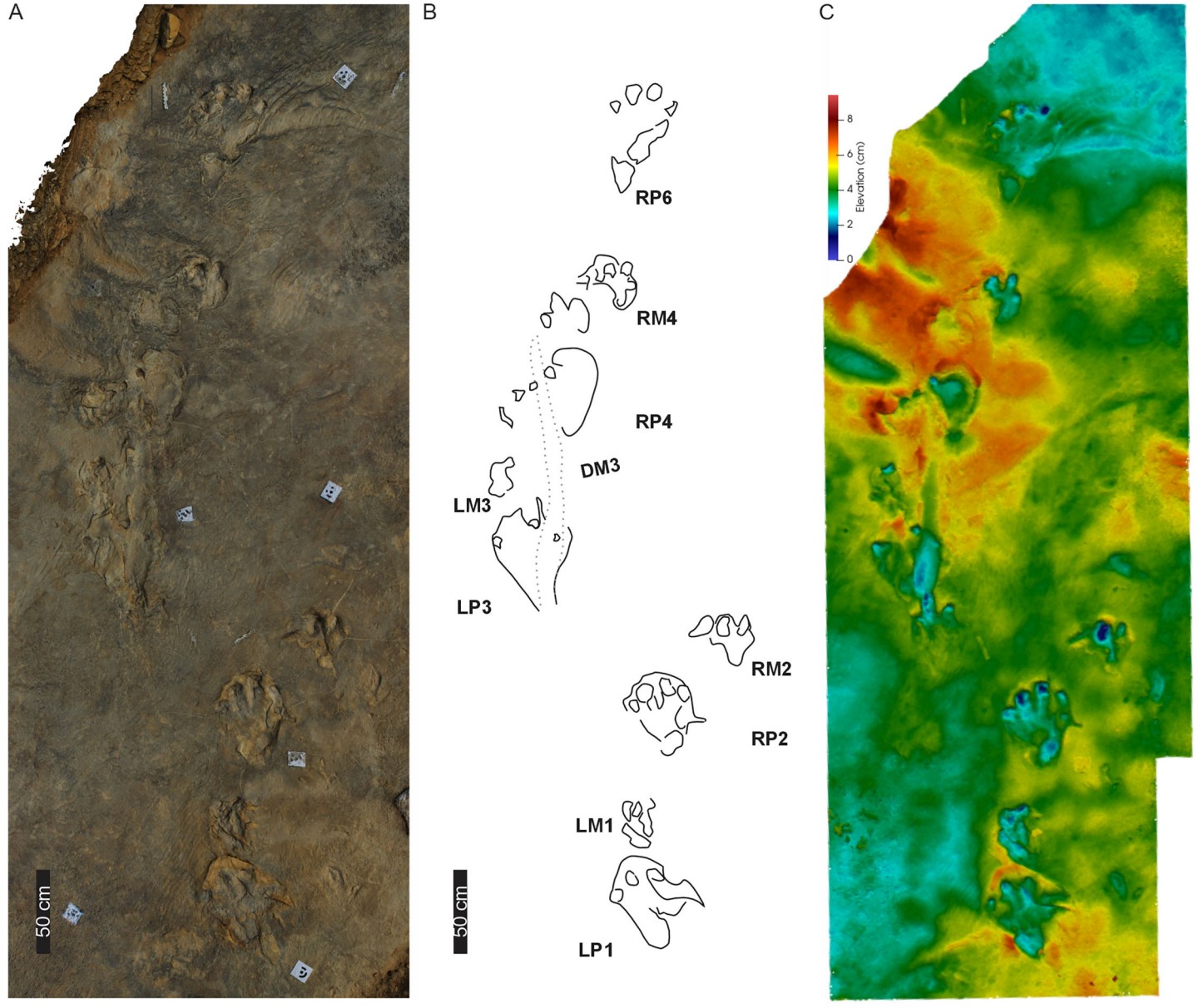

**Figure 6 Trackway PT2 at the Phuthiatsana track site.** (A) Orthophotograph, (B) interpretative and labelled outline drawing, and (C) false-colour depth map of PT2 (depth increases with darker greens). Pes-manus pairs 3 and 4 are associated with a medial drag trace (DM3; dashed line; see Fig. S2 for more detail). For location, see Fig. 2B.                    

footfall pattern (pace and stride length av. 130 and 247 cm, respectively; Table 1) but the interpes and intermanus distance are comparable (av. 130 and 131 cm, respectively). PT2-LP5 is absent and PT2-RP6 shows divergence from the main trajectory of the trackway indicating possible shift in the trackmaker's direction. Pes pace angulation ranges between 133–138°, whereas the manus pace angulation av. 97°. PT2 has a narrow trackway width (av. 85 cm, 65% PTR; Table S1). The estimated $h$ is c. 2.0 m, GAD is 1.43 m, and the gait ratio is 1.24, with an estimated walking speed of 5.7 km/h (Tables 1, S1).

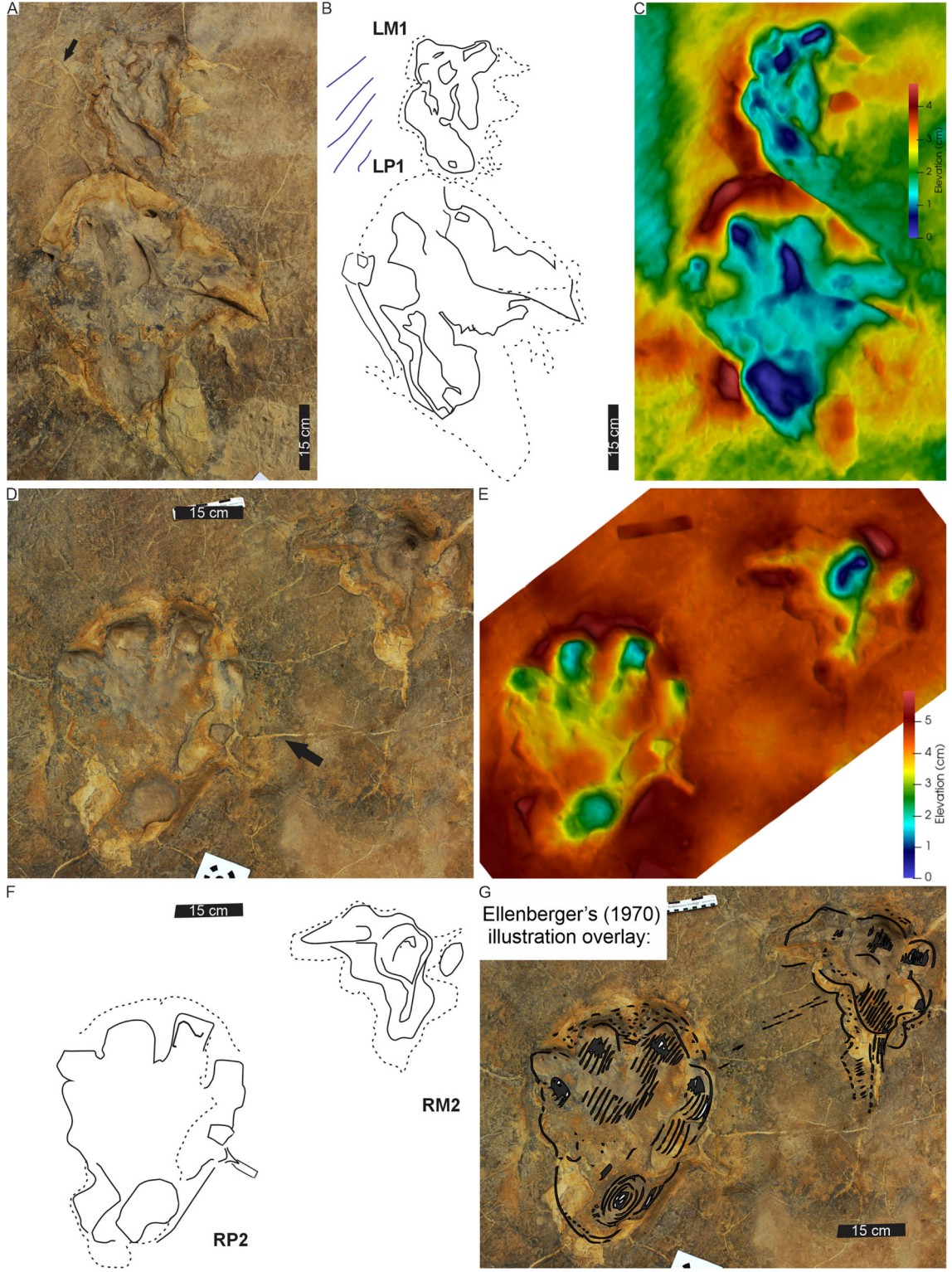

**Figure 7 Pes-manus pairs in trackway PT2.** (A) Orthophotograph, (B) interpretative outline, and (C) false-colour depth map of the first left pes-manus pair (PT2 LP1–LM1). (D) Orthophotograph, (E) interpretative outline, and (F) False-colour depth map of the second right pes-manus pair (PT2 RP2–RM2 ). (G) Same as in (B) but overlain by *Ellenberger (1970)* original interpretative outline. Black arrows in (B) and (F) point to desiccation cracks, blue lines indicate ripple mark crests, and dashed lines indicate soft sediment disruption.

**Drag trace:** PT2 is associated with a single broad (c. 15–18 cm wide), U-shaped, medial, and oblong-shaped drag trace, DM3 (Figs. 6, S2). It is c. 180 cm in length and partially obscures tracks PT2-LP3-RP4 (Figs. 6, S2). DM3 is shallow (c. 1 cm deep) and has smooth surface ornamentation and expulsion rims. It follows a straight path, shallowing and tapering towards its easterly end.

**Substrate consistency and impression depth**: Substrate saturation was heterogeneous along the length of PT2 and is demonstrated by the highly variable plastic deformation displayed by the track morphologies and associated sedimentary structures (*e.g.*, interference ripple marks). These features indicated water pooling on the surface during track maker movement. Water run-off features forming shallow, narrow channels, and rivulets (Figs. 6, S2A′) crosscut and further modified the tracks (*e.g.*, PT2-RP4 and the drag trace DM3; Figs. 6, 7B). In all cases, pes and manus tracks are associated with a marginal sediment ridge in front of the trend of the digits due to sediment deformation and displacement. When displaying plastic deformation, as in PT2-LP1 (Fig. 7A), marginal upfolds of sediment carried over the pes digits and "heel" partially infill this track, conversely, PT2-RP2 shows the morphology of the pes and manus with less modification associated with appendage-substrate interaction. The pliable substrate illustrates the weight-bearing portions of pes and manus in PT2, and the depth of the manus impressions is equal to and occasionally deeper than those of the pes, with the maximum load carried, largely, through the digits.

## PT3 trackway

PT3's tetradactyl pes tracks are asymmetrical and moderately large (PL c. 46 cm, PW av. 36 cm wide; av. PL/PW ratio of 1.3; Figs. 8, 9; Table 1). The pes traces are preserved as four blunt, rounded phalangeal tips, with digit I more oblong than digits II–IV and digits II–IV forming a distinct arc shape. Digit I is c. 6 cm shorter than the other digits. A basal, oblong to oval "heel" impression is present at the rear, external margin of the track, almost directly below digit III (Figs. 8, 9). Digit divarication is c. 51° (originally recorded as I^II: 15°–22°; I^IV: 42°–49° in *Ellenberger, 1972*). Previously recorded U-shaped hypices are weathered away.

The trackway PT3 is composed of seven consecutive footprints and runs east westerly semi-perpendicular to PT1 and PT2 (Figs. 2B, 8, 9). A manus is not originally reported for this trackway, however, in the field, four rounded digit-like impressions (width c. 25 cm) are associated with the first right and possibly the third right pes trace in PT3 (indicated in Fig. 8). In the latter, a possible sediment push-back ridge is noted posteriorly to the trace (Fig. 9B″). The stride and pace (as measured from tops of digit III) average 164 and 116 cm, respectively. PT3 trackway width averages 90 cm with the pace angulation averaging 131°. The estimated hip height (h) of the bipedal trackmaker is c. 1.8 m, and gait ratio is 0.9 with a walking speed of 3.1 km/h (Tables 1, S1).

**Drag trace:** A sinuous, U-shaped, discontinuous drag trace (DM4), also noted by *Ellenberger (1972)*, runs along the midline of PT3 partially obscuring footprints 5 and 6 (Fig. 8). The c. 141 cm long DM4 is an unornamented groove tapering at both ends, with a

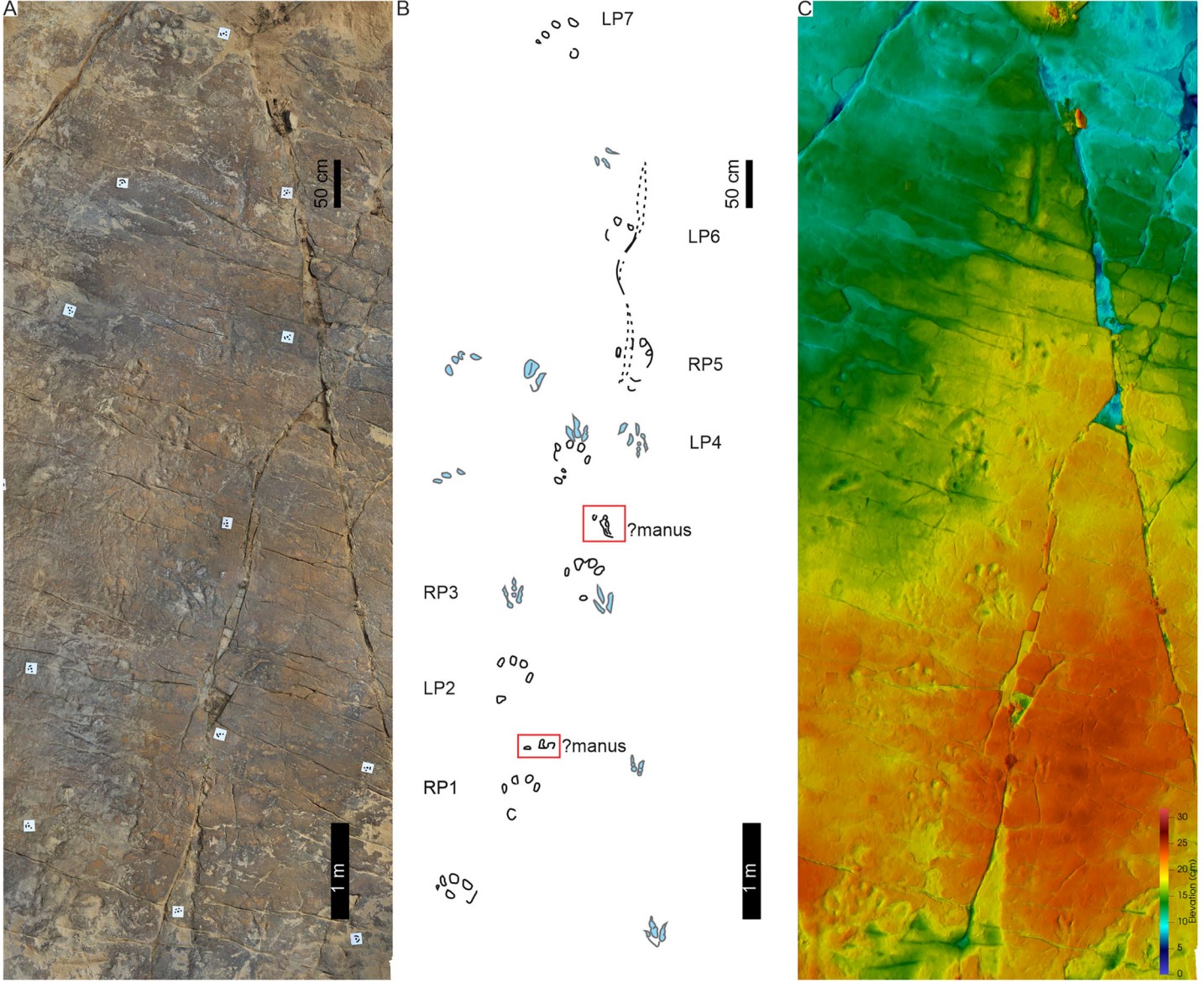

**Figure 8 Trackway PT3 at the Phuthiatsana track site.** (A) Orthophotograph, (B) interpretative outline, and (C) false-colour depth map of bipedal trackway PT3 (depth increases with cooler colours). In (B), note potential manus impressions (red boxes) in front of consecutive right pes tracks; a straight to sinuous drag trace (DM4; black dashed outline) from track RP5 past track LP6; and other ?tetradactyl and tridactyl tracks (blue). For more details, see Fig. 9. For location, see Fig. 2B.

maximum width of 8 cm and c. 1.5 cm deep. In placement and morphology, DM4 is similar to DM3 (Figs. 6, S2).

**Substrate consistency and impression depth:** Ripple marks and tridactyl tracks preserving distinct phalangeal pads and claw traces (Fig. 8) are suggestive of a homogenously saturated substrate across this part of the exposed palaeosurface. The original description of PT3 confirms a similar morphological preservation as the tridactyl tracks (*i.e.*, phalangeal pads noted in *Ellenberger, 1972*) but since their first

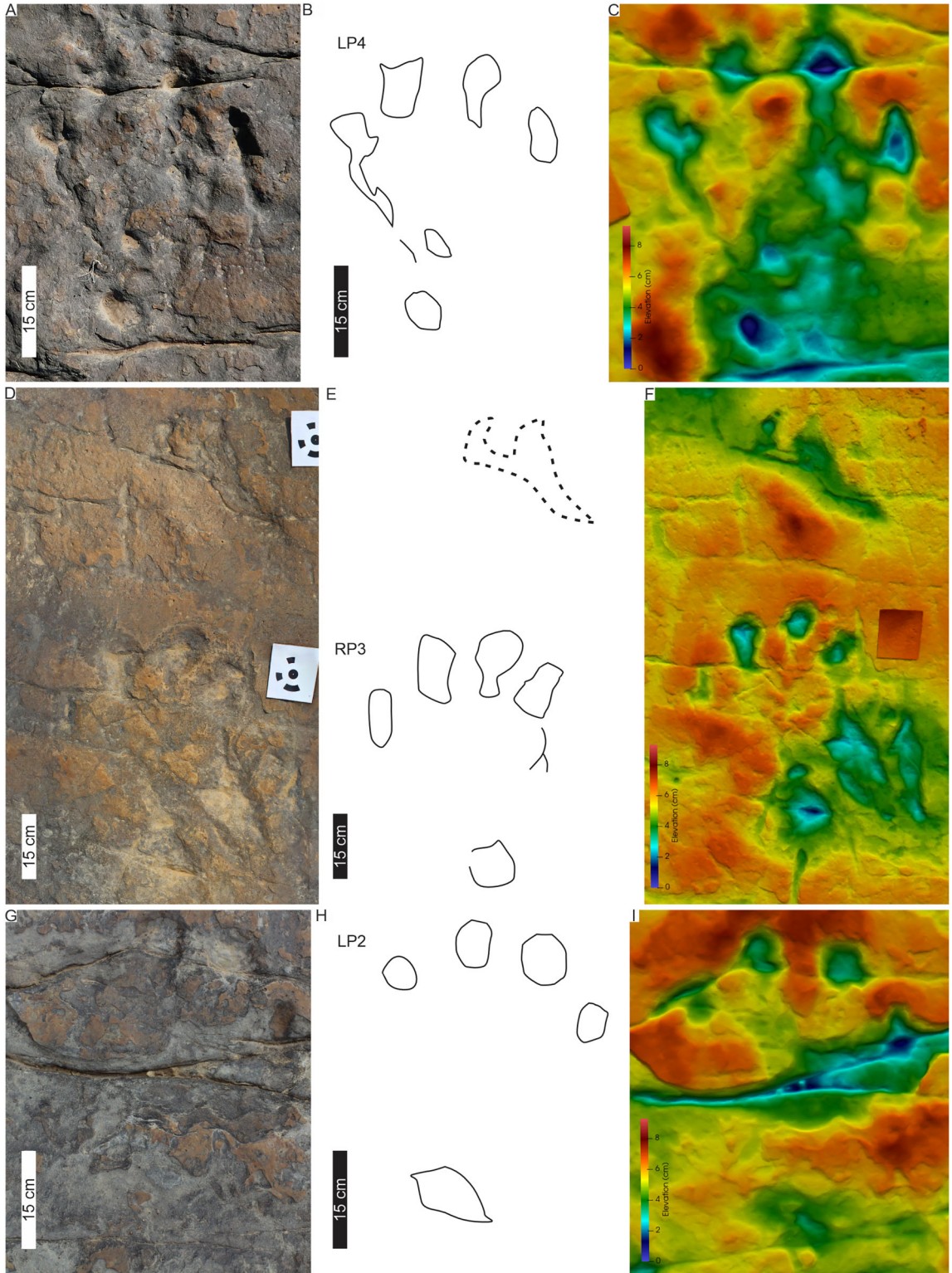

**Figure 9 PT3 pes tracks.** Each track (A–C, LP4; D–F, RP3 and G–I, LP2) is illustrated with three different images: Orthophotograph (A, D, G), Interpretative outline (B, E, G) and false-colour depth map (C, F, I). In (B′), note overprinting by both a tridactyl footprint and the potential manus impression (dashed outline) with push-back ridge highlighted in (B″).

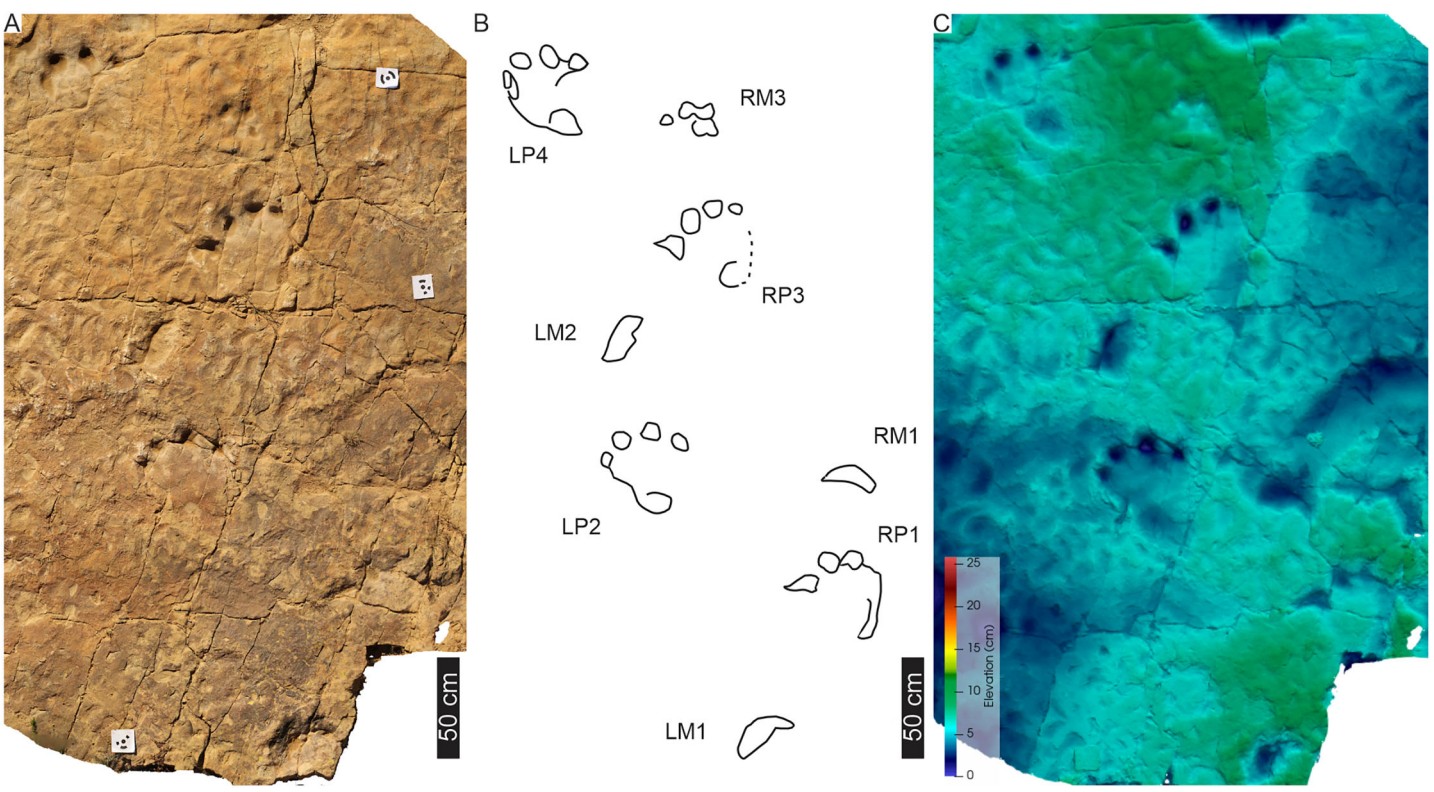

**Figure 10  Trackway MP1 at the Maphutseng track site.** (A) Orthophotograph, (B) interpretative outline drawing, and (C) false-colour depth map (depth increases with cooler colours) of four pes-manus pairs and an isolated manus track (LM1). For location of MP1, see Fig. 2B.

description in the 1960s, weathering has significantly reduced the morphological integrity of the more shallowly impressed morphological features.

## MAPHUTSENG TETRADACTYL TRACKWAY (<215.4 MA)

Based on our repeated investigation of the Maphutseng track surface, the tetradactyl trackway described herein as MP1 (Fig. 10) has never been documented before. In MP1, the pes tracks are large (av. PL of 45 cm, av. PW of 41 cm, with surface area of c. 1,024 cm$^2$), asymmetrical, tetradactyl tracks, slightly longer than wide (av. PL/PW of 1.05 ± 0.05), and semi-digitigrade to semi-plantigrade. The digits are deeply impressed, distinct from the raised sole, and have subrounded distal morphologies, with digit I occasionally preserving a distinct medially oriented, sickle-shaped claw mark. Digit III is the longest, with digits II and IV being the shortest and subequal in length (I = IV < II < III). The total sole impression is elongate to lunate in shape, giving the impression of a medial longitudinal arch. For three of the pes tracks, an oval "heel" impression at the rear of the sole is observed and may either represent a large digit IV pad or the digital metatarsal region. For most of the pes impressions, this "heel" morphology is not very clear but in track MP1-LP4 it is longer than wide. Other than this feature, metatarsal-phalangeal pads are absent in MP1 pes impressions.

Most of the manus impressions preserve low anatomical fidelity, simply having an open arc morphology that is wider than long (av. ML of 12.5 cm, av. MW of 27.5 cm, av. ML/MW of 0.46; Fig. 10; Table 1). Manus MP1-RM4 preserves the most detail with four discernable clawless digit impressions. The manus tracks are anterolateral and turned outwards relative to their corresponding pes tracks.

Overall, MP1 comprises three manus-pes pairs, an isolated pes and an isolated manus, with strong heteropody. Pes morphologies are more defined than manus morphologies. The straight trackway is 276 cm long (measured from the "heel" of pes 1 to the "heel" of pes 4) and 104 cm wide, with a slight outward rotation of the manus impressions. The pes trackway ratio (PTR) is 41% suggesting a "medium gauge". The pace length and pace angulation for the pes and manus impression are 127 cm and 99°, and 118 cm and 113°, respectively, with manus tracks impressed 40–45 cm away from their pes counterpart (measured from pes digit IV to the outer-side of the arc manus). The estimated $h$ is c. 1.81 m with a GAD of 1.38 m. The calculated gait is 1.14, indicative of a walking speed of 4.1 km/h (Table 1).

**Drag trace:** None observed.

**Substrate consistency and impression depth:** Water saturation of the substrate along MP1 was homogenous and wet, as the surface is still covered in very well-formed ripple marks without any desiccation cracks around or in the tetradactyl tracks. The northern end of the track-bearing surface does show evidence for desiccation (on the palaeosurface and within larger tridactyl tracks). Relative to other tracks on the palaeosurface, these large tetradactyl tracks are moderately deeply impressed, with the toes, and to a lesser degree the "heels", having consistent depths along the trackway. Tridactyl tracks on the surface are impressed more shallowly and preserve additional morphological features such as phalangeal pads and claw traces.

## HA FALATSA-LIBATAOLONG TETRADACTYL TRACKS (<209.6 MA)

### Trackway HF1

HF1 has moderately large, asymmetrical pes (av. 45 cm PL, 41 cm PW; av. PL/PW ratio 1.1, surface area is c. 1,118 cm$^2$; Figs. 11, 12; Table 1) defined by, at least, three prominent claw traces (digits I–III). This gives the appearance of a semi-digitigrade pes with a plantigrade trace of digit IV. Claw traces curved medially and are anteromedially oriented with respect to the direction of travel (Figs. 11, 12). Digit I is the shortest but with a significant medially orientated claw trace (av. 13 cm long; claw-free digit I length: total digit I length ratio: 0.4). Digits II–IV orient forward, with the claw trace recurving strongly medially. The pes' postero-lateral margin is pronounced along digit IV and shows a convex margin. The ungual impressions free lengths average 11, 9, and 6 cm for digits II, III, and IV, respectively. Both digits I and II bear claw traces slightly separate (I^II: 19°–21°) and splayed relative to digits III and IV (≤ c. 10°). The external lateral margin of digit IV (and V) forms a narrow indentation that runs the length of the posterior margin

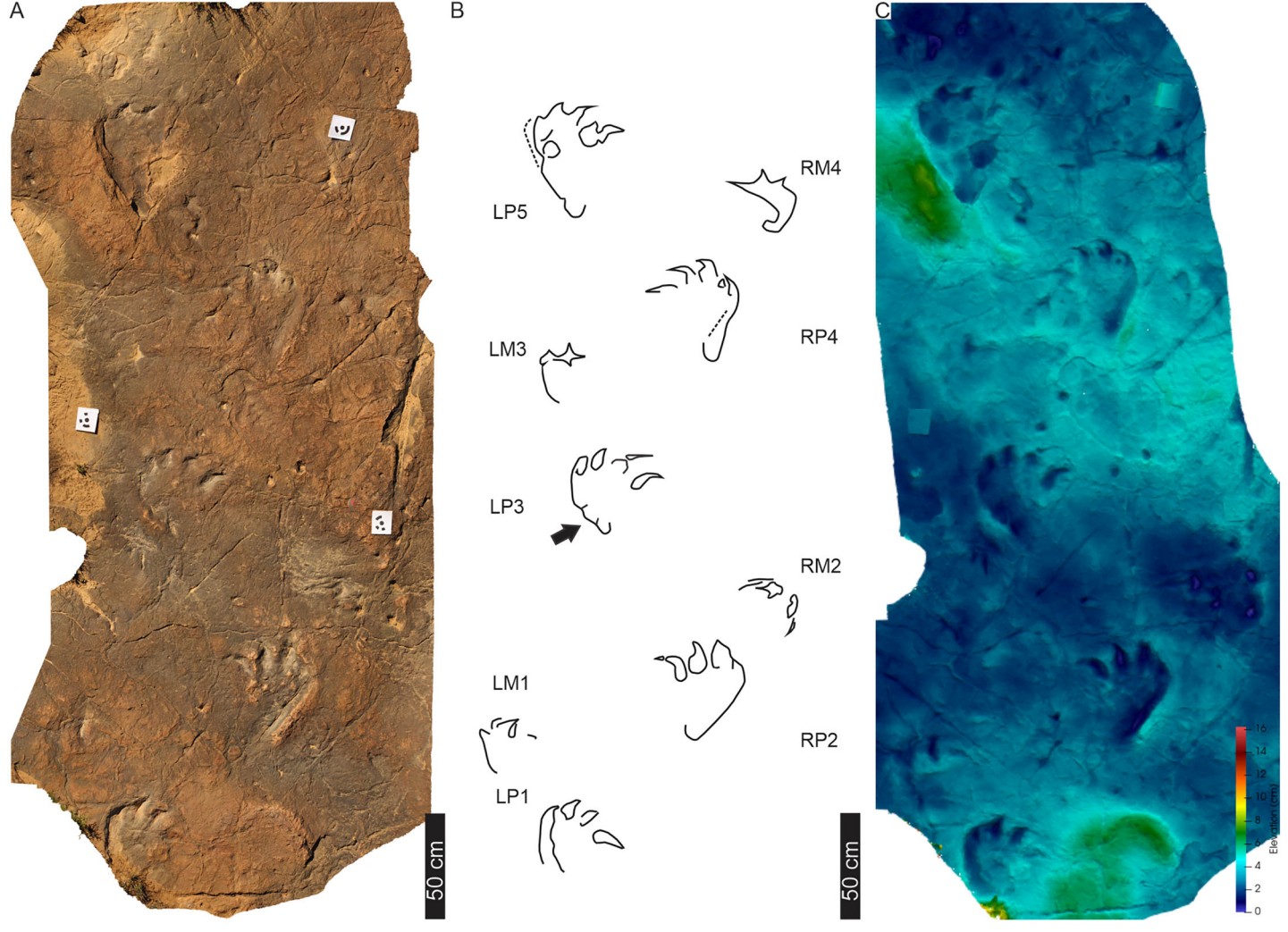

**Figure 11 Trackway HF1 at the Ha Falatsa track site.** (A) Orthophotograph, (B) interpretative outline, and (C) false-colour depth map (depth increases with darker blue hues) of HF1 four pes-manus pairs and a fifth pes track (LP5). Black arrow indicates *Otozoum*-like digit IV segmentation. The trackway is *Tetrasauropus unguiferus* by *Ellenberger (1972)*. For more details, see Figs. 12 and 13. For location, see Fig. 2B.

travel (Figs. 11, 12). Divarication between pes digits I–IV is ≤ av. 56° and hypices are U-shaped. Metatarsophalangeal pads are not observed (Fig. 12).

The HF1 manus tracks (Fig. 12) show poor field preservation relative to the casts taken several decades ago (see fig. 5 in *D'Orazi Porchetti & Nicosia, 2007*). *In-situ* manus impressions are longer than wide (av. 30 cm ML, c. 21 cm MW; av. ML/MW ratio 1.4), tetra-to ?tridactyl with strongly medially oriented claw trace of digit I (Figs. 11, 12). Claw traces on digits II and III have pointed anterior margins (*e.g.*, HF1-RM4; Fig. 11). Digits show wide, U-shaped hypicies. Manus placement is consistently external and anterior to the pes with c. 40–50 cm (between pes digit III and manus digit III). The manus is pronated, usually placed in line with the subsequent pes step and the manus axis is parallel to the trackway and pes midlines. Manus measurements by *Ellenberger (1972)* record a length and width of 32 cm with digit III projecting 0.4 cm and a total divarication of 50°

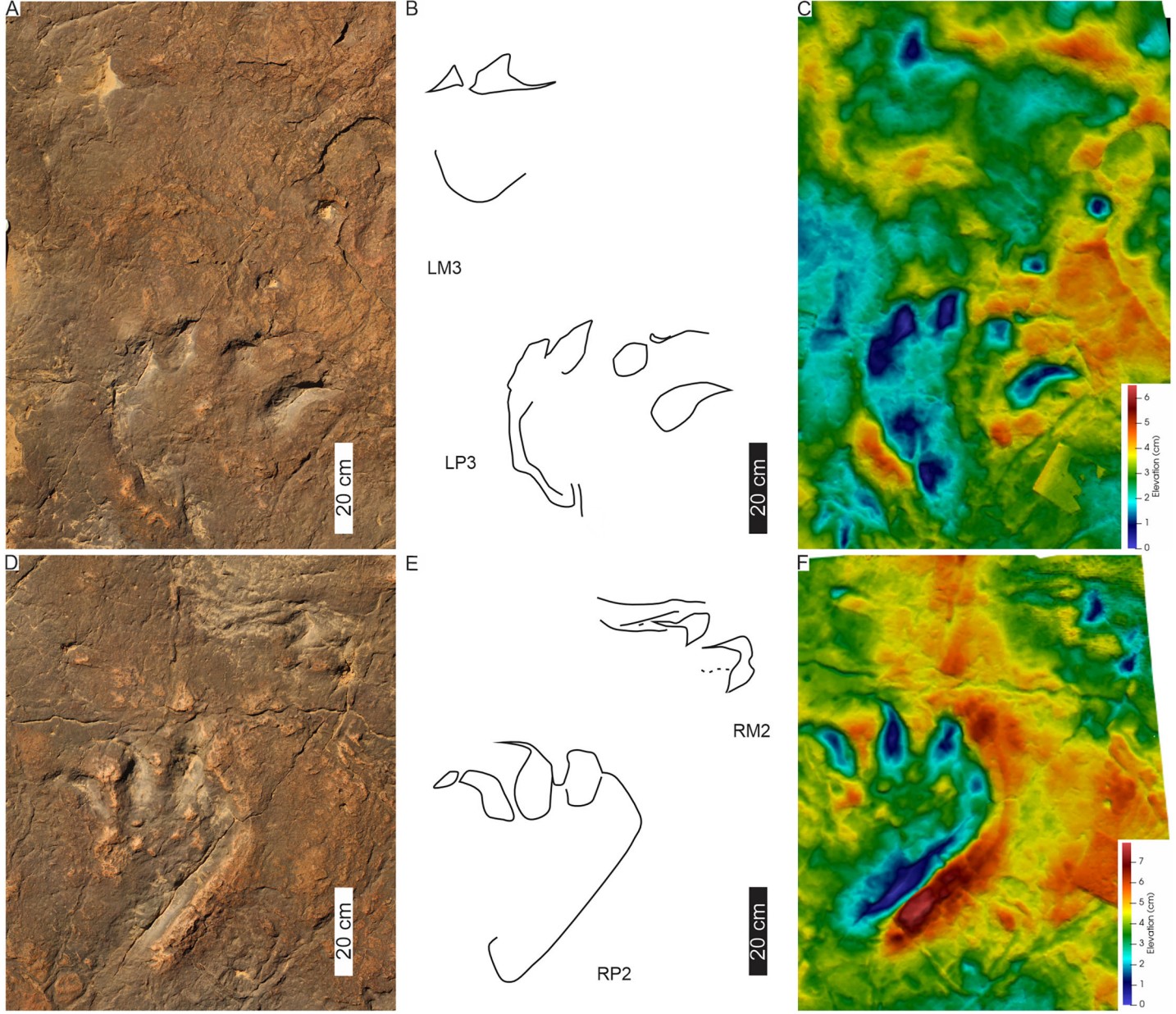

**Figure 12 Pes-manus pairs in trackway HF1.** (A) Orthophotograph, (B) interpretative outline, and (C) false-colour depth map of the third left pes-manus pair (RP3–RM3). (D) Orthophotograph, (E) interpretative outline and (F) false-colour depth map of the second right pes-manus pair (RP2–RM2). Due to weathering, the impression of the pes-manus pairs are shallower compared to the original description, casts and photographs of *Ellenberger (1972)*.

(herein, ML, MW and manus angulation of 21, 30 cm, and 81°, respectively). The greatest impression depth is within the digit and claw traces (Fig. 12).

The c. 4 m long main HF1 trackway is composed of five consecutive pes and four manus traces (Fig. 11) with additional tracks and an associated drag trace present in an adjacent sloped palaeosurface (Figs. 13A, 13B) that project and curve seamlessly onto the main HF1 track layer. On the main surface, the long axis of both the manus and pes tracks are roughly parallel to the trackway midline. The intermanus distance is larger than the interpes

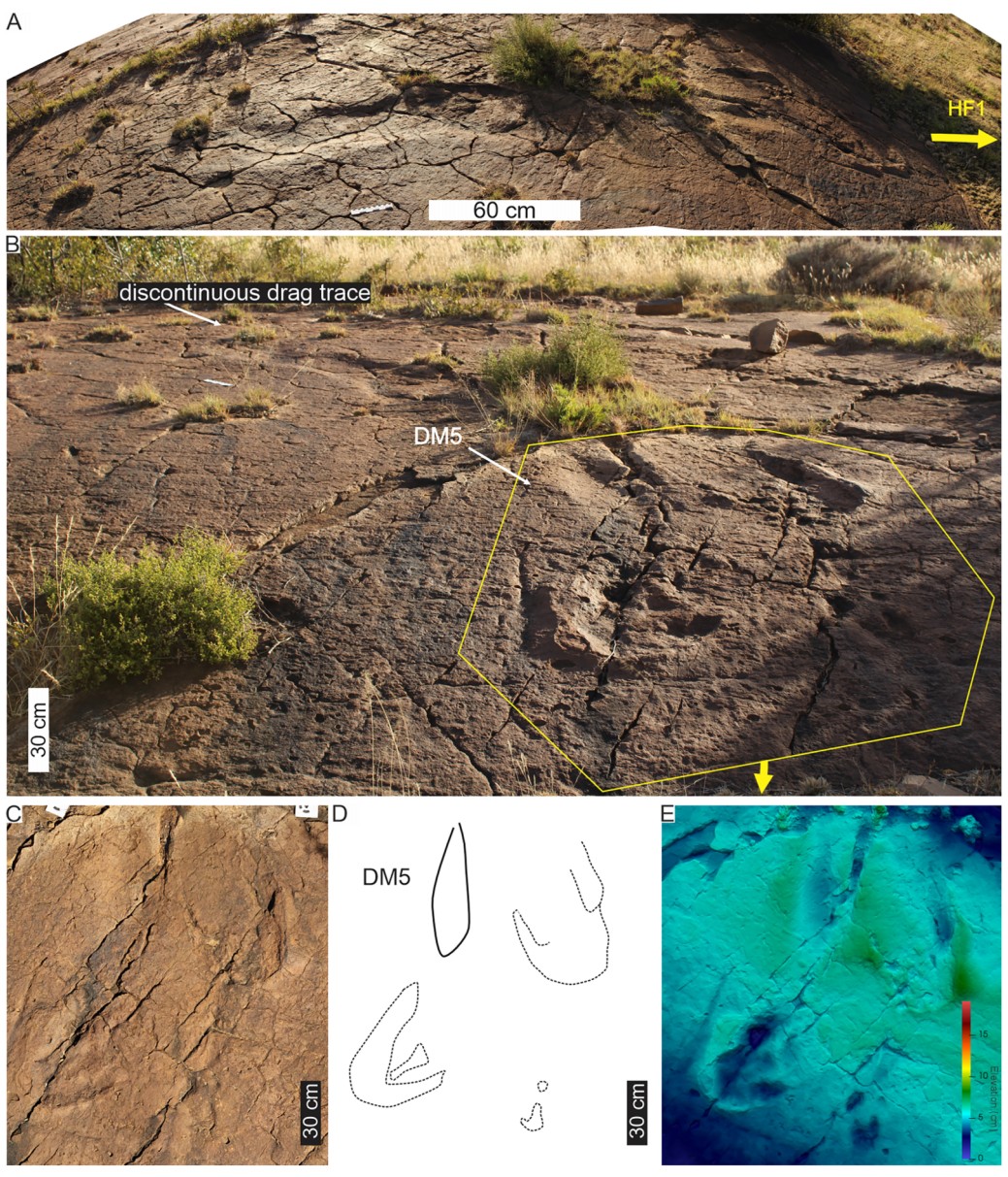

**Figure 13 Linear drag trace (DM5) at Ha Falatsa track site.** (A) Palaeoslope and drag traces associated with HF1. Yellow arrow indicates direction to main HF1 palaeosurface shown in Fig. 12. (B) Extent of drag traces on palaeosurface that terminate in the recorded DM5 and the associated and deformed pes tracks on the sloping surface directly leading into (yellow arrow) the flat part of trackway HF1 (Fig. 12). (C–E) Photograph, line drawing and depth map of tracks (dashed lines) associated with the drag trace (solid line; DM5).

distance (153 and 112 cm, respectively) with the average pes and manus angulation of 116° and 81°, respectively. The trackway width is 92 cm and HF1 has a PTR of 46% ("medium gauge"). The estimated $h$ is c. 1.7 m, GAD is 1.3 m, and the gait is 1.1 with a walking speed of 4.3 km/h (Tables 1, S1). Tracks in HF1 are the well-known ichnogenus *Tetrasauropus unguiferus* (*e.g.*, *Ellenberger, 1970*, *1972*; *Ellenberger, Ellenberger & Ginsburg, 1970*).

**Drag trace:** HF1 preserves a linear drag trace (Fig. 13; terminates at DM5; Fig. 13C) that is ~120 cm long, c. 30 cm wide (but variable along length), >2 cm deep, U-shaped groove, and laced between strides (Figs. 13B, 13C). It is associated with plastically deformed pes impressions and a palaeoslope.

**Substrate consistency and impression depth:** Soft-sediment deformation in and around HF1 tracks is evidenced by sediment ridges directed medially and laterally around the external margin of pes digit IV and the claw trace associated with pes digit I (Figs. 12B, 13C). The pes tracks are strongly entaxonic with robust digit I traces, but with the maximum depth occurring along the posterior margin of the pes (along digit IV) and within the digits and their claw traces. Manus claw traces are penetrative whereas pes claw impressions lie parallel to the sediment surface. The modified tracks and related sedimentary structures of the HF1 trackway (Fig. 12) and the associated palaeoslope (and drag trace DM5; Fig. 13) illustrate greater plastic deformation of the substrate on the palaeoslope (Figs. 13A, 13B). Weathering of HF1 has obliterated much of the morphological details noted in *Ellenberger (1970, 1972)* and *Ellenberger, Ellenberger & Ginsburg (1970)*.

## Trackways LB1 and LB2

LB1 and LB2 pes tracks are moderately large (av. 40 cm PL, av. 33 cm PW; av. PL/PW ratio 1.4; Tables 1, S1), asymmetrical (Fig. 14) with the track axis parallel to the trackway midline. The four long, rounded and separated digit impressions are anteriorly facing with respect to the direction of travel (Fig. 14). Digit III is the longest and projects c. 5 cm past II–IV, digit I is the shortest. Preservation of the digital pad traces is imperfect from the current material (Fig. 14), although *Ellenberger (1972)* recorded a phalangeal formula of 2-3-4-5-1. Divarication between digits I–IV is 61° (I^II: 25°; II^III: 23°; III^IV: 13° in *Ellenberger, 1972*, p. 58). Generally, claw traces are absent, although *Ellenberger (1972)* describes both large blunt claw impressions and in LB2 sharp claw traces that penetrated the substrate (*e.g.*, Fig. 14E). The "heel" is an oblong to oval impression on the rear, external margin of the track. This "heel" is c. 20–27 cm below the tip of digit IV and appears to be made by the impression of digit V. *Ellenberger (1972)* notes that this triangular 'pedicel' is a unique identifying characteristic of these tracks (*Ellenberger, 1972*, p. 59). No manus is recorded.

The now obliterated Libataolong track-bearing surface contained two trackways of bipedal, tetradactyl animals, herein referred to as LB1 and LB2. LB1 and LB2 comprised six and three footprints, respectively. Using a scaled drawings of LB1 (Fig. 14A′), we confirm the trackway measurements of *Ellenberger (1972)* as follows: the stride and pace average 116 and 71 cm, respectively; trackway width is c. 61 cm (narrow trackway gauge; Table 1); and pace angulation is relatively low at 109° (stride: 1.48 m; pace: 0.74 m, pes angulation: 27–28° in *Ellenberger, 1972*, p. 59). The estimated hip height of the bipedal trackmaker is c. 1.6 m, gait is 0.7 with a walking speed of 2.1 km/h (Table 1).

**Drag trace:** None observed.

**Substrate consistency and impression depth:** Given the loss of this palaeosurface, we rely on the original descriptions that indicate the substrate's water saturation along LB1

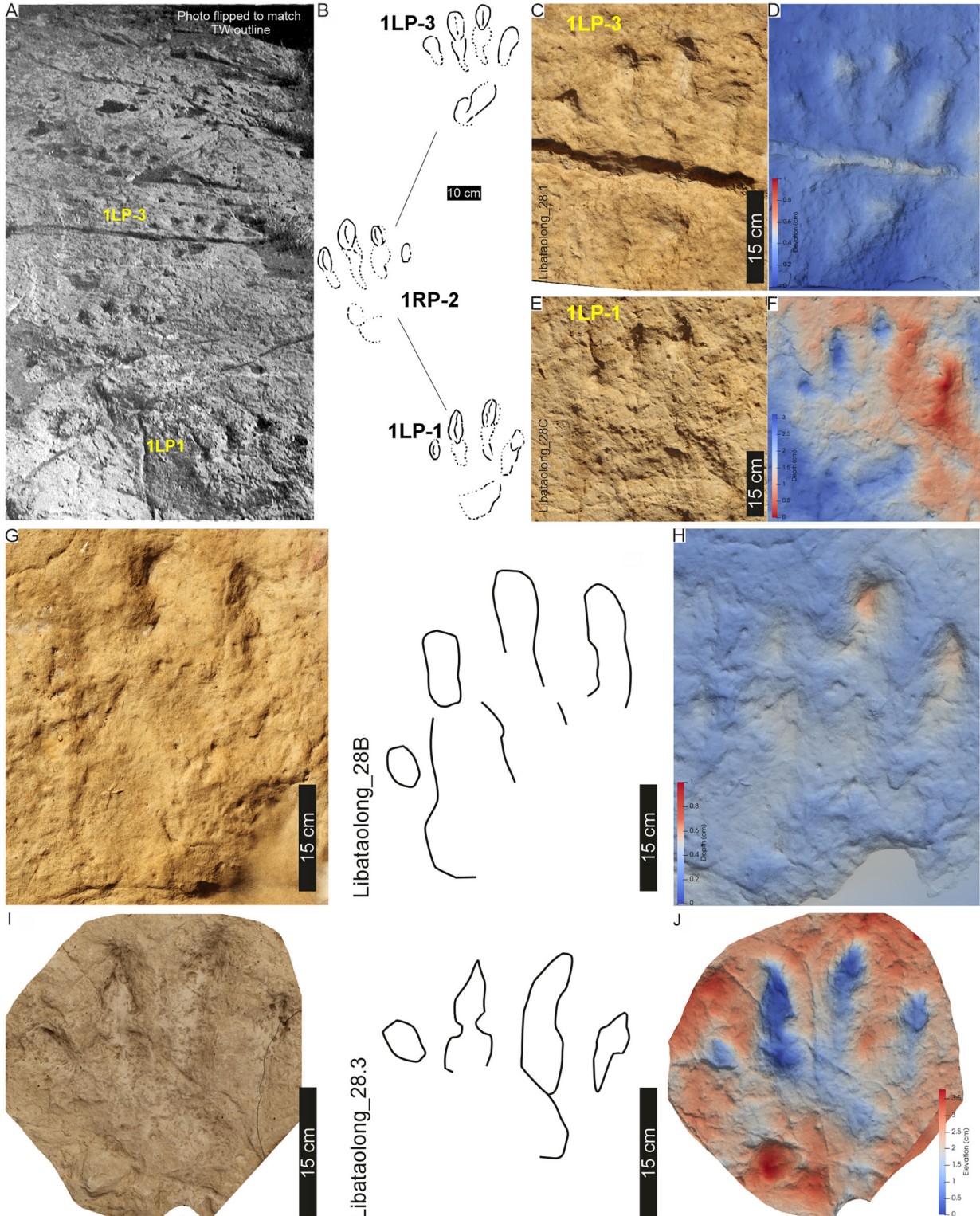

**Figure 14 Tracks and trackway LB1 at the Libataolong track site.** (A) Photograph of LB1 from *Ellenberger (1972*, his plate V), (B) Interpretative outline of LB1 from *Ellenberger & Ellenberger (1958*, trackway K, originally). (C–J) Orthophotographs and (B′–E′) False-colour depth maps of four tracks at this site. Tracks (C–F) are from LB1 and tracks (G–J) are from LB2. Both trackways were made by bipedal animals, and the tracks were interpreted to be *Pseudotetrasauropus bipedoida* by *Ellenberger (1970, 1972)*. For location, see Fig. 2B.

and LB2 was heterogenous. This is shown by the relatively deeper tracks of LB1 that contrast the more shallowly imprinted, cross-cutting tracks of LB2 (*Ellenberger, 1972*). The latter were, nonetheless, described with elongated and penetrative claw impressions (Fig. 14E). LB1 and LB2 were initially suggested to be undertracks (*Ellenberger & Ellenberger, 1958*: p. 65), however this may be a function of the "sandy beach with clay pebbles" (*Ellenberger, 1972*, p. 58) onto which the tracks were impressed. Later, *Ellenberger (1972*, p. 58) describes phalangeal pads and claw traces cross-cutting desiccation cracks from the same surface. Other sedimentary structures (*e.g.*, ripple marks) are not apparent in the photograph of the trackway (*Ellenberger, 1972*: plate V) nor the parts of the plaster of Paris replicas that extend beyond the tracks.

## DISCUSSION

### Ichnotaxonomy

The main hypothesis, on studying the *in situ* material, is that these lEF tracks represent *Pseudotetrasauropus* or *Tetrasauropus* and are distinct from *Otozoum*, as previously exerted (*D'Orazi Porchetti & Nicosia, 2007*). We find that these lEF tetradactyl tracks (Figs. 3–14; Table 1) share the following common pes features: (i) digit traces II–IV have blunt distal ends (bar HF1); (ii) digit III is the longest, digit I is shorter (c. 5 cm) than digits II–IV; (iii) the long axis of pes tracks are roughly parallel to the trackway midline; (iv) the posterior "heel" pad is at the external margin of pes and c. 27–30 cm below the tip of digit IV; (v) a plantar sole impression; (vi) the interpes distance displays low variability and pes pace angulation are comparable. Quadrupedal forms are represented by PT1, PT2, MP1 and HFI with tetradactyl manus impressions regularly impressed and showing degrees of pronation. Only in trackway HF1 are manus claw traces noticeable, and the manus is consecutively external to the pes track.

### *Pseudotetrasauropus* Ellenberger, 1972

As per the emended diagnosis of *Pseudotetrasauropus Ellenberger, 1972* (*D'Orazi Porchetti & Nicosia, 2007*), the following pes tracks herein are ascribed to the ichnogenus: PT1 (Figs. 3, 4), PT2 (Figs. 6, 7), PT3 (Figs. 8, 9), MP1 (Fig. 10), LB1 and LB2 (Fig. 14). Previously, PT2, PT3, LB1, and LB2 trackways were attributed to *Pseudotetrasauropus* by *Ellenberger (1972)* and represent both quadrupedal (PT2: *P. jaquesi*) and bipedal (PT3: *P. augustus*; LB1, LB2: *P. bipedoida*) forms.

### Ichnospecies *Pseudotetrasauropus bipedoida* Ellenberger, 1972

Description: Based on tracks LB1, LB2 and PT3, *P. bipedoida* represents a moderately large (40–46 cm), longer than wide (PL:PW of 1.2–1.3) pes, with the digits separated along their length (I^IV of 51°–57°). *P. bipedoida* is a bipedal trackway displaying a narrow-gauge (TW:PW of 1.8–2.5) and with a pace angulation of 109–131° and SL: PL of 2.9–3.5.

Casts of tracks LB1 and LB2 represent the holotype *Pseudotetrasauropus bipedoida* from trackways A and B of Ellenberger (for LB1, see trackway 'K' in *Ellenberger & Ellenberger, 1958*: p. 66; *Ellenberger, 1972*: plates V and VII). PT3 is identical to "trackway H" of *Ellenberger et al. (1963*: p. 316) and subsequently formally described as

*Pseudotetrasauropus augustus* by *Ellenberger (1972*, planche X) as "showing several anatomical details: phalangeal pads, sole, heel, variability in finger spacing, tail mark". Another *P. augustus* trackway, with tail trace, is recorded on the same surface but was not studied herein (*e.g.*, *Ellenberger, 1972*, planche XI) due to its weathering (preservation grade of 0–1; cf. *Belvedere & Farlow, 2016*; grade 0.5–1.0; *Marchetti et al., 2019*). This likely represents the "Phuthiatsana 5XY 1" cast (Figs. S1L–S1N), which contains an associated manus-pes trace as well as a tridactyl track. This cast is housed at the Morija Museum and Archives in Lesotho (without a number).

Trackways LB1, LB2 (*P. bipedoida*), and PT3 (*P. augustus*) are herein considered to be synonymous, as was previously denoted in *D'Orazi Porchetti & Nicosia (2007)*. *Ellenberger (1972)* cited several differences between the two ichnospecies, particularly referencing the triangular pedal sole impression in *P. bipedoida* as "spécifique chez ceite spèce", and a function of the metatarsals III and IV impressions (*Ellenberger, 1972*). The latter is considered by *D'Orazi Porchetti & Nicosia (2007)* as the impression of digit V. Herein, we consider that the "heel" impression in *P. bipedoida* (Fig. 14) and *P. augustus* (PT3; Fig. 9C) to be similar (present on external margin, elongate and falling below digits III, IV), and likely the impression of digit V. It is important to note that the *P. bipedoida* trackway illustration provided by *D'Orazi Porchetti & Nicosia (2007*; their fig. 7, p. 232) from *Ellenberger et al. (1963*; his trackway H, p. 316) is, in fact, '*P. augustus*' and may have influenced their description, but it also highlights the marked similarities. Together, these insignificant morphological differences are likely a function of substrate and do not have bearing on the establishment of a new ichnospecies.

## Ichnospecies *Pseudotetrasauropus jaquesi* Ellenberger, 1972

**Holotype replica:** Plaster of Paris replicas of tracks labelled "Phuthiatsana 34B" (10.17602/M2/M526635), "Phuthiatsana 35" (10.17602/M2/M526627) housed as unnumbered casts at the Morija Museum and Archives, Lesotho and MorphoSource models (DOI 10.17602/M2/M526717; 10.17602/M2/M526681).

**Type locality:** lower Elliot Formation, Stormberg Group (middle Norian). GPS: 29°21′30.02″S, 27°36′35.61″E.

**Revised diagnosis:** *P. jaquesi* represents a quadrupedal trackway showing defined heteropody. The pes are large (45–56 cm), longer than wide (PL:PW of 1.0–1.2), with digits arranged in a fan (I^IV of 55°–63°) and displaying separation along distal margin transitioning into a plantar pad with deeper "heel" impression. The latter is an oval, basal pad trace present below digits II–IV along the lateral margin. Pes digit I has a medially oriented claw trace. Manus impressions have a length: width ratio of 0.4–0.5 and lack claw impressions *P. jaquesi* is a narrow-gauge trackway (TW:PW of 1.5–2.4) with a wide range in pace angulation (99–138°) and ratios in stride length: PL of 3.9–5.0. The pes orientation is roughly parallel to marginally inwardly rotated towards the trackway midline. Manus orientation and placement, relative to the trackway midline, is variable. Trackways show comparable interpes and intermanus distance. *P. jaquesi* differs from *P. bipedoida* in the following features: quadrupedal; pes less elongate (PL:PW ratio of 1.0–1.2 compared to

1.2–1.3 in *P. bipedoida*) and averaging 13.7% larger; wider range in pace angulation (99 to –138° *vs.* 104–132°) and SL:PL ratios (3.9–5.0 *vs.* 2.9–3.5 in *P. bipedoida*, respectively).

**Remarks:** *D'Orazi Porchetti & Nicosia (2007)* excluded *P. jaquesi* from the ichnogenus *Pseudotetrasauropus* based on the presence of the manus and referred it to *Lavinipes Avanzini, Leonardi & Mietto, 2003*. However, although larger in size, pes in PT1, PT2, and MP1 (Figs. 4, 6, 10) follow the diagnosis for *Pseudotetrasauropus* and other differences (*i.e.*, splay of the digits, claw trace on digit I) are a likely function of the pliable substrate coupled with subsequent weathering. These three trackways are from the same (*i.e.*, PT1, PT2) or similarly aged track-bearing palaeosurfaces (Fig. 2), and their significant difference is their quadrupedality. The similarity in the pes traces is likely compounded by the probable basal sauropodomorph trackmaker given the general parallels in the (poorly preserved) pedal anatomy of these animals. Trackway PT1 was not described by *Ellenberger et al. (1963)* or *Ellenberger (1970, 1972)* and does not match the bipedal *Pseudotetrasauropus acutunguis* (defined by *Ellenberger, 1970, 1972*) at Phuthiatsana. PT2 was previously described as *Tetrasauropus jaquesi* (*Ellenberger, 1970*), *Pseudotetrasauropus jaquesi* (*Ellenberger, 1972*, planche XI), and is currently synonymized with *Lavinipes* (*Avanzini, Leonardi & Mietto, 2003*; Early Jurassic, Italy) as ?*Lavinipes jaquesi* in *D'Orazi Porchetti & Nicosia (2007)*. We refute the latter based on our descriptions of the original material, which *D'Orazi Porchetti & Nicosia (2007)* were unable to investigate, and reaffirm its placement within *Pseudotetrasauropus*. MP1 was not previously assigned to *Pseudotetrasauropus* as it is first described herein.

Given morphological similarities between the *Pseudotetrasauropus* pes tracks of bipedal and quadrupedal forms, and no significant variation in manus morphology, it is practical to group these trackways under *Pseudotetrasauropus*. However, it raises the question as to the practicalities of an ichnogenus with bipedal and quadrupedal forms, and if trackmaker behavior plays a role in the presentation of a bipedal or quadrupedal stance (*i.e.*, bipedal PT3 with possible manus impressions; Fig. 8). Altogether, we assert that *Pseudotetrasauropus* is a unique ichnogenus, and we ascribe all the lEF material reviewed to this ichnogenus, except for HF1 (discussed below). *P. bipedoida* is used with bipedal trackways, and quadrupedal trackways (*e.g.*, PT1, PT2, MP1) are placed within *P. jaquesi* given its priority and in consideration of *D'Orazi Porchetti & Nicosia (2007)*. Other differences between these tracks, besides the presence of a manus, are controlled by substrate conditions and subsequent weathering.

### *Tetrasauropus Ellenberger, 1972*

HF1 is markedly different to and c. 10 Ma younger than the tetradactyl tracks of Phuthiatsana and Maphutseng. HF1 is Ellenberger's (1972) *Tetrasauropus unguiferus* and describes a relatively large quadruped with a pronated manus. The manus is held out and away from the pes and trackway midline and has distinctive medially directed claw impressions on at least three digits. The sole of the foot, as in PT1 and MP1, likely bore a fleshy sole pad and there is no clear distinction of the pedal digits bar the distal ungual. *D'Orazi Porchetti & Nicosia (2007)* consider *Tetrasauropus* to be a valid ichnogenus and,

based on our assessment of the trackway, we reaffirm the emended diagnosis of *Tetrasauropus unguiferus* *Ellenberger, 1972* provided by the cast material.

### Ichnospecies *Tetrasauropus unguiferus* *Ellenberger, 1972*

Description: *T. unguiferus* (HF1; Figs. 11, 12) is characterized by: (i) four robust, medially-directed, and separated ungual impressions on the pes and, to some extent, the manus (at least on digit I), although the pes itself does not show inward rotation; (ii) weak heteropody (unlike PT1, MP1 and PT2, manual impressions in HF1 are larger; *i.e.*, the ratio of ML:PL between PT1, MP1, PT2, and HF1 as 1:5, 1:4, 1:4 and 1:2, respectively); (iii) manus tracks are anteriorly and externally place relative to pes; (iv) pes are entaxonic: larger ungual digit I and consecutively smaller digits II–IV; and, (v) no constricted heel-region, as in *Pseudotetrasauropus* pes tracks, but pronounced digit IV trace along its length towards the posterior portion of the foot.

### OPEK plexus

All OPEK members display consistent linear relationships between pes length, width, and trackway width with medial rotation of pes and the concave inward curvature of digit and claw traces (*Lockley et al., 2023*). These are defined as "otozoid" characters of the prosauropod foot and contrast with the non-OPEK tracks made by derived sauropod trackmakers (*e.g.*, *Eosauropus*; *Sander & Lallensack, 2018*; *Lockley et al., 2023*). Given the morphology of *Tetrasauropus*, it fits within the "otozoid" morphologies and can be considered within the OPEK plexus, as has been recently discussed by *Lockley et al. (2023)*. However, both *Pseudotetrasauropus* and *Tetrasauropus* show deviation from some "otozoid" characters. Specifically, *Pseudotetrasauropus* has straight, forward orientated digits, with the except of digit I and its claw trace in the quadrupedal morphotype *P. jaquesi* and, similarly, *Tetrasauropus* only displays medial curvature of ungual traces and the external margin of digit IV. Furthermore, using the long axis of the pes, through digit III, neither ichnogenus shows pronounced inward or outward rotation of the pes, and this may be an important trackway feature. Lastly, the difference between *P. jaquesi* and *Tetrasauropus* manus placement and orientation may be related to the forelimb posture.

In comparison to OPEK-member, *Otozoum*, *Pseudotetrasauropus* trackways herein are differentiated by several, substrate unrelated, features: (i) pedal digits do not curve medially; (ii) digit III is the longest digit; (iii) *Pseudotetrasauropus* digit IV is shorter than digit III by 11% (av. c. 5 cm). In comparison, digit IV is longer by 6% (av. 1.7 cm, $n = 7$) in bipedal *Otozoum moodii* specimens WUM182, WUM 183, and trackway WUM680, and 5.5% (av. 2.5 cm, $n = 7$) shorter in the only quadrupedal *Otozoum* trackway 1AAR1 (measurements taken from *Farlow et al., 2022* for WUM specimens, and *Masrour & Pérez Lorente, 2014* [their fig. 3] for 1AAR1); (iv) digits II–IV show a degree of separation along their lengths, but particularly when impressed in pliable substrates (*e.g.*, PT2-RP1, PT3-LP5); and, (v) digits II–IV end in blunt rectangular-shaped traces (*e.g.*, Fig. 6B) that are prominent in both firm and plastic substrates. In PT2, particularly, these may give the impression of keratinized unguals (?hooves as in hippopotamus) as opposed to keratinized claws terminating the digits, however, LB digits II–IV may terminate in pointed margins

**Table 2  Morphometric measurements for OPEK plexus members.**

|  | PL (cm) | PL/PW | ML/MW | Pes Pace/Stride | Stride/PL | TW (cm) | TW:PW |
|---|---|---|---|---|---|---|---|
| *Otozoum* isp. | 36 | 1.2 | 0.7 | 0.5 | 4.1 | 48.9 | 1.6 |
| *Pseudotetrasauropus* isp. | 47 | 1.2 | 0.5 | 0.6 | 3.9 | 84.4 | 2.1 |
| *Evazoum* isp. | 14 | 1.1 | no manus | 0.6 | 5.3 | 21.4 | 2.1 |
| *Kalosauropus* isp. | 13 | 1.4 | no manus | 0.5 | 5.2 | 12.7 | 1.4 |
| *Tetrasauropus* isp. | 45 | 1.1 | 1.4 | 0.6 | 4.2 | 92.0 | 2.2 |

Note:
Data for *Otozoum*, *Evazoum* and *Kalosauropus* summarized from *Lockley et al. (2023)*.

(*e.g.*, Libataolong 28.3; Fig. 14). Altogether, these contrast with the key features of *Otozoum* described by *Rainforth (2003)* and recently discussed in *Farlow et al. (2022)*: four stout pedal digits (often with distinct digital pad traces) curving medially; digit III longest although digit IV can be the longest (*e.g.*, WUM182, WUM680); length of each digit pronounced as hypexes are deeply situated; digit V impression large and impressed below the fused metatarsophalangeal pad trace of digits III and IV; metatarsophalangeal impressions below digits I, II and III–IV; elongate "heel" mark below digit V.

Remaining OPEK-members, the Late Triassic *Evazoum* (*Nicosia & Loi, 2003*) and Early Jurassic *Kalosauropus* (*Ellenberger, 1970*; *Mukaddam et al., 2021*), are not comparable with *Pseudotetrasauropus* and *Tetrasauropus*. *Evazoum* and *Kalosauropus* are relatively small (<15 cm av. PL), may show functionally didactyl and tridactyl footprints with the former having a distinctly well-developed, oval, digit II proximal phalangeal pad. Another large, Late Triassic quadruped *Eosauropus* (non-OPEK) is dissimilar to *Pseudotetrasauropus* and *Tetrasauropus* based on the orientation of the pes and its digits (*Lockley, Lucas & Hunt, 2006a*; *Lockley et al., 2023*; Table 2).

Lastly, we reinforce *Lockley & Meyer (2000)*'s consideration that *Pseudotetrasauropus* (and *Tetrasauropus*) is a distinct ichnotaxon distinguishable from *Brachychirotherium*. In both lEF ichnotaxa, pes are digitigrade, digit IV is not the longest and does not curve outwards as might be expected in a non-dinosaurian archosaur. And while the basal "heel" pad in *Pseudotetrasauropus*, often referenced as a digit V impression, is similar in shape (and less so in orientation relative to digits) to *Brachychirotherium* isp. (*e.g.*, *Klein, Lucas & Haubold, 2006*), other aspects of the pes (digit divarication, orientation, almost subequal lengths of digit II and IV) and the trackway pattern, including manus placement, are not comparable (see *Lockley et al., 2023*).

## Basal sauropodomorph manus placement and its relation to forelimb condition

From the body fossil record, a primitive forelimb in most basal sauropodomorphs is, generally, laterally flexed (crouched), with minimal extension at the glenoid, a shallow radial fossa, flexion at elbow, and a semi-supinated hand with gently arced metacarpus (*Otero, 2018*). A typical basal sauropodomorph manus consists of five digits with digits I–III being long, robust, and clawed, and digits IV–V are reduced, laterally orientated with

blunt unguals. Digit I is reduced relative to II and III and has a pollux. Functionally, grasping ability and limited pronation is feasible but when coupled with the shortened length of the forelimb, restricted rotation under loading and limited pronation, its use in quadrupedal locomotion is problematic (*Mallison, 2010*; *Reiss & Mallison, 2014*). In this case, contact with the ground would be specific to arrested periods of feeding/drinking whereby the digits would be orientated laterally and the palm medially in a digitigrade posture (*Reiss & Mallison, 2014*). This orientation has been noted in the quadrupedal instance of *Otozoum moodii* (*Rainforth, 2003*), although *Lockley et al. (2023)* illustrate that is not strictly the case (see their figs. 5A, 5C, 5E, p. 9). Given the rarity of sauropodomorph manus impressions, it is important to showcase these lEF tracks as they may inform ichnogeneric assignment (*Lockley et al., 1994*), forelimb condition, and behavior.

Morphologically, *P. jaquesi* and *T. unguiferus* manus traces are distinguishable with *T. unguiferus* having a tetradactyl manus with medial deflected claw traces, a pronated orientation, and a partial palm impression. Additionally, weak heteropody defines *Tetrasauropus* (ML/MW of 1.4) in contrast to *Pseudotetrasauropus* (ML/MW ratio of 0.4–0.5, respectively). In terms of trackway parameters, the intermanus distance is larger in *Tetrasauropus* and is always external and anterior to the pes, compared with *Pseudotetrasauropus*, where the manus is generally orientated parallel and proximal to the hindfoot. The former implies a crouched quadrupedal stance, whereas the latter establishes a more parasagittal posture, but with degrees of variability that may be substrate and/or behaviorally controlled.

In comparison with other sauropodomorph trackways, the placement and orientation of *Tetrasauropus* manus tracks (*i.e.*, consistently pronated with the palmar trace directed caudally) assumes restricted flexed forelimb mobility during the locomotion of the trackmaker. Could a fleshy palmar pad trace in *Tetrasauropus*, when coupled with the external manus placement, suggest cushioning to minimize forces operating on the forelimb held in a wider, flexed stance in a larger-bodied animal during movement? Contrastingly, *Pseudotetrasauropus* quadruped trackmakers exhibit a higher degree of craniolateral movement in the forelimb and pronation in the manus traces due to their variable placement and orientations. This is further showcased by the drag traces (DM1, DM2; Fig. 5) discussed below (*e.g.*, PT1-RM1 *vs.* PT1-RM3 or PT2-LM1). Additionally, the strong heteropody and clawless digits of *Pseudotetrasauropus* manus tracks are reminiscent of a sauropod-like manus track with reduced phalanges or unguals held in a flexed posterolateral position. Speculatively, extramorphological features in manus PT2-RM2 and PT2-RM4 (superficially) may reflect flexed phalanges (and hyperextension occurring at the metacarpophalangeal joint—see also *Thulborn, 1990*, p. 283). Furthermore, evidence for either a non-uniform locomotory suite or facultative quadrupedality may be illustrated in the irregularity of *P. jaquesi* intermanus distance, forelimb footfall pattern and manus orientation within and between trackways. This is reinforced by the inconsistency in the morphology of the manus tracks coupled with the presence of extramorphological features.

## Drag traces

Drag traces of possible digit(s) and body parts represent the more intriguing aspects of these trackways (Figs. 3, 7, 8, 13), and represent valuable information about limb movements (*Gatesy & Falkingham, 2017*). From our track sites, water-saturated substrate conditions, palaeo-terrain topography (*e.g.*, DM5, Fig. 13), and other alterations (*e.g.*, trackway course change in PT2 and at HF1) play a role(s) in the registration of these traces with evidence of departure from 'normal' posture. HF1, PT2 and PT3 share similar-looking broad, U-shaped, medially orientated traces (*e.g.*, DM3, DM4, DM5) that are enigmatic but may represent a portion of the body (*e.g.*, ?tail), given their morphology and orientation. Tail traces are rarely reported associated with sauropodomorph-related ichnotaxa (*Kim & Lockley, 2013*) and none have been associated with quadrupedal trackways. For example, a single tail trace is identified for the bipedal trackway of *Otozoum* (V-shaped cross-section and 'herringbone striae'; specimen WU 725; *Rainforth, 2003*, her fig. 5e).

Contrastingly, the PT1 drag traces (DM1, DM2; Fig. 5) are thin medially curving traces that originate at the external margin of the manus PT1-RM2 and end at the base of the proceeding pes track (PT1-RP4; Fig. 5). DM1 and DM2 may represent the impression of the tail (*e.g.*, crocodile tail drag trace; *Milàn & Hedegaard, 2010*, their fig. 9), whereby the swing of the body and tail (*e.g.*, *Dickinson et al., 2000*) during rapid movement, produces a medially directed trace and would imply a more sprawling limb posture. However, given that the trackway evidence supports parasagittal locomotion with extension and flexion of the limbs, and in the forelimb from the glenoid (*Van Buren & Bonnan, 2013*), DM1 and DM2 are not tail traces. It is probable, given their origin, that DM1 and DM2 represent the movement of the manus, particularly as it subdivides into two drag traces, potentially indicative of traces of digits. These medially arcing drag traces originate at PT1-RM2 manus, which shows a palm-backwards orientation (Figs. 3, 4A) and this degree of pronation is unusual. Thus, supination of the manus from a pronated position could have produced an inwardly arcing trace with the separate drag traces a function of the external digits encountering the substrate. It also describes the arc the autopod takes during locomotion that is not typical in (bipedal) sauropodomorphs (*i.e.*, an arc outward rotation/craniolaterally during the initial phase of limb protraction; *Thulborn, 1990*, his fig. 6.23, p. 261; *Langer, 2003*).

## Underfoot weight distribution pattern

Here, the pes and manus trace depth and associated soft-sediment deformation have been used as proxies for the distribution of weight and the final pressure distribution across the foot/manus (*Falkingham et al., 2011*; *Falkingham, Bates & Mannion, 2012*; *Strickson et al., 2020*). We assume morphological variations caused by trackmaker speed are negligible because all estimated trackway speeds are walking gaits (Table 1). We conceded that differential substrate conditions and associated sediment movement can obscure and enhance the underfoot pressures during a trackmaker's walk.

A noticeable feature for all these lEF trackways, through time, is the trace of a moderately large sole pad (plantar pad) reflected in elevated depth-pressure patterns.

It likely assisted with weight distribution during locomotion and is conspicuous in quadrupedal trackways (*i.e.*, the elongated digital traces are absent, cf. PT1 *vs.* LB1; Figs. 3 *vs.* 12). During weight bearing stances, a soft tissue pad would reduce the stress and pressure acting on the limbs in larger animals (*Jannel, Salisbury & Panagiotopoulou, 2022*). Hypothetically, this implies that a somewhat upraised skeletal pedal morphotype (*e.g.*, see *Plateosaurus engelhardti* in *Jannel, Salisbury & Panagiotopoulou, 2022*) with a soft tissue pad below digits II–IV is possible in lEF sauropodomorphs and reflected in their tracks.

Trackways of *Pseudotetrasauropus* have consistent depth-pressure patterns between bipedal and quadrupedal forms (Figs. 4, 7, 10, 11). The ungual phalangeal traces of digits I–IV and the probable digit V pad ("heel") show the greatest variation in depth-pressure pattern. Conversely, the midfoot region shows a depth-pressure pattern possibly associated with and explained by a fatty pad that distributed pressures across this region more evenly (plantar pad discussed above; *Panagiotopoulou et al., 2016*; *Panagiotopoulou, Pataky & Hutchinson, 2019*). During the stance phase in quadrupedal *Pseudotetrasauropus* pes tracks, sediment marginal ridges and expulsion rims (*e.g.*, Figs. 4–6), theoretically, illustrate the heel-to-toe movement of the foot as it rolled off the midsole fatty pad and the active role of the digits II–IV during the push-off/kick-off phase of each step cycle. The latter is represented by the sediment ridge (*e.g.*, PT1 Fig. 3) that can overprint any posteriorly displaced sediment ridge of the manus impression. In more saturated substrates, the obliquely positioned (relative to substrate plane) pes digit I forms a broad, sickle-shaped depression and an indirect soft sediment deformation feature (informally labelled as a sediment 'expulsion flap' in Figs. 4A, 7A; also see *Gatesy, 2003*). This deformation exaggerates the size and shape of digit I traces, highlighting its separation from digits II–IV (shown by a sediment rise), and illustrates its mobility relative to the other digits.

In general, *Pseudotetrasauropus* manus tracks present no clear consistency in their depth patterns and this may be controlled by the shifting of weight depending on forelimb placement. The posteriorly displaced sediment ridge in several manus tracks (*e.g.*, PT1, PT2; Figs. 3–6), although smaller than that produced by the pes, suggests digits backwardly pushed sediment during propulsion. The difference in manus: pes surface areas in *Pseudotetrasauropus* suggests weight being born by the hindfoot because of their larger surface areas, regardless of negligible depth variance between the hind- and forefeet.

In *Tetrasauropus* (Figs. 13, 14), in contrast to *Pseudotetrasauropus*, the margin of pes (along digit IV) shows the greatest depth and likely highest pressures associated with weight-bearing during the stance phase. A push-up/expulsion ridge parallel to the digit IV trench trace may indicate both the transfer of weight and the motion of the pes during touch down to kick-off. The weight distribution on the outer margins of the pes is contra the entaxony expected in sauropods (*Bonnan, 2003*; *Wright, 2005*; *Lallensack et al., 2017*). Relative to the depth-pressure pattern in the pes, the *Tetrasauropus* manus are less deeply impressed. The pes: manus surface areas are more comparable than in *Pseudotetrasauropus* (low heteropody) with the manus trace reflecting a fleshier autopod. *Tetrasauropus* manual and pedal tracks provide evidence for fleshy feet that likely served to

dispense the load away from individual digits and claws (*Strickson et al., 2020*) and support a large animal.

## Potential track makers

*Pseudotetrasauropus* and *Tetrasauropus* trackmakers have been repeatedly considered to be "prosauropods" by Ellenberger and co-workers (*e.g.*, *Ellenberger & Ellenberger, 1956a*, *1956b*; *Ellenberger, Ellenberger & Ginsburg, 1970*) and this has been reaffirmed more recently (*e.g.*, *D'Orazi Porchetti & Nicosia, 2007*; *Lockley et al., 2023*). *Charig, Attridge & Crompton (1965)* specifically pointed out the 'Blikana dinosaur' (*Blikanasaurus* cromptoni) as an apt (bipedal) "Cinderella"-candidate for *Pseudotetrasauropus* at Phuthiatsana and Libataolong. *Ellenberger (1972)* suggested that the Maphutseng dinosaur, now *Kholumolumo ellenbergerorum* (*Peyre de Fabrègues & Allain, 2019*), corresponded well with *Pseudotetrasauropus*. Refining these general attributions and using pedal digit lengths, claw morphology and bipedal gait, *D'Orazi Porchetti & Nicosia (2007)* considered a *Plateosaurus*-like non-sauropod sauropodomorph as the most likely candidate for *Pseudotetrasauropus*.

From a stratigraphic perspective, *Kholumolumo* is the only body fossil directly associated with a tracksite (underlies the Maphutseng tracksite by c. 15 m; *Ellenberger, 1972*, p. 79). Currently, *Melanorosaurus* (NMQR 1551) and *Plateosauravus* are lowermost lEF trackmakers with limited biostratigraphic ranges. Other lEF sauropodomorphs, *Eucnemesaurus*, *Sefapanosaurus*, and *Meroktenos* have longer (but presently uncertain) ranges and only *Blikanasaurus* could be a candidate trackmaker at all the tracksites (*McPhee et al., 2017*; *Viglietti et al., 2020*). Caution is taken in trying to "match" the ancient echoes of the fleshy manus/pes in soft sediment and fossil bones. This is complicated by the lEF trackways representing taxa yet to be documented in the body fossil record and/or the stratigraphic overlap between better age-constrained ichnotaxa and body fossil taxa with less dependable provenance data (*e.g.*, *McPhee et al., 2017*). While the lEF basal sauropodomorph dinosaur body fossil record is rich, it also lacks articulated or complete manus or pes and makes further associations between the trackway and fossil records difficult.

Post-cranial remains preserving fossil pes are recorded for *Blikanasaurus* (left pes SAM K403; *Galton & Van Heerden, 1985*, *1998*), *Kholumolumo* (complete right pes; NMQR1705; *Krupandan, 2019*, his fig. 27; *Peyre de Fabrègues & Allain, 2019*) and *Eucnemesaurus entaxonis* (right pes; BP/1/6234; *McPhee et al., 2015b*, their fig. 16). *Melanorosaurus* and *Sefapanosaurus* (BP/1/386; *Otero et al., 2015*) are currently the only other lEF sauropodomorph with partial/incomplete pes. Southern African basal sauropodomorphs' pes had a phalangeal formula of 2-3-4-5-1 and four unguals across the pes (1-1-1-1-0; *Galton & van Heerden, 1998*) with digit III the longest, digit I bearing a pronounced claw and a vestigial digit V. Moreover, the body fossil evidence (*e.g.*, *McPhee et al., 2017*) suggests two pedal morphotypes from the available materials: (i) a stout, robust form (Fig. 15A; *Blikanasaurus*), and (ii) a slim, elongate and more 'prosauropod' form (Figs. 15B, 15c; *e.g.*, *Kholumolumo*, *Melanorosaurus* NMQR 1551).

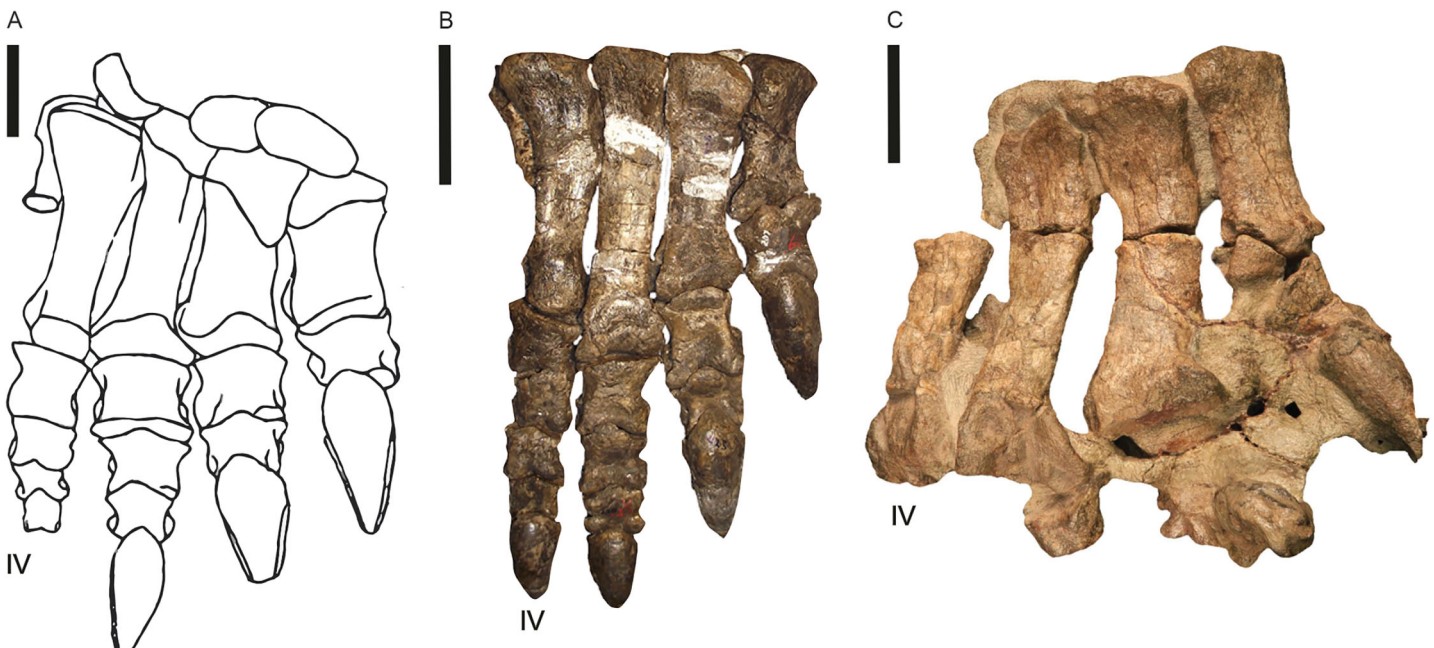

**Figure 15 Pes of three lower Elliot Formation sauropodomorphs.** (A) *Blikanasaurus cromptoni* (holotype SAM K403; from *Galton & Van Heerden, 1985*). (B) *Kholumolumo ellenbergerorum* (NMQR1705). Reverse image in anterior view from *Krupandan (2019)*. (C) *Eucnemesaurus entaxonis* (BP/1/6234). Right pes in dorsal view taken from *McPhee et al. (2015b)*. Digit IV is indicated. Scales are 5 cm.

The slim pes morphotype corresponds to both potentially quadrupedal and bipedal track makers in *Melanorosaurus* (NMQR 1551; ?facultative quadrupedal due to their relatively large humerus/femur ratios) and *Kholumolumo* (lEF's largest ?bipedal non-sauropodan sauropodomorphs; *Peyre de Fabrègues & Allain, 2019*). In *Kholumolumo*, bipedality is assumed based on the proportions of its forelimb and manus with distinctive digit I. *Kholumolumo* has a pedal anatomy (Fig. 15B) that convincingly fits the *Pseudotetrasauropus* "slipper", and its fossil remains are stratigraphically closely associated with *Pseudotetrasauropus*-bearing tracksites. It is plausible as a candidate trackmaker as already suggested by *Ellenberger (1972)* but for different trackways on the Maphutseng track-bearing surface.

Contrastingly, the stouter, robust pes of *Blikanasaurus* is speculated to be a heavier facultative biped that would favor a supportive, robust entaxonic pedal morphology—particularly if regularly rearing during foraging (*Galton & van Heerden, 1998*; *McPhee et al., 2015a*, *2017*). The limited but distinctly short and robust remains of *Blikanasaurus* keep its larger anatomy obscured and the animal has, therefore, been described as both bipedal and quadrupedal (*Charig, Attridge & Crompton, 1965*; *Galton & van Heerden, 1998*; *Yates, 2008*). It is noteworthy that robust morphologies (limb and pedal bone structures) and shortened distal limb segments are often linked to the increase in body size and the shifting of the center of mass to a more graviportial mode (*Carrano, 2005*; *Lockley, Kukihara & Mitchell, 2008*). Moreover, in *Blikanasaurus* the distal tarsals are displaced

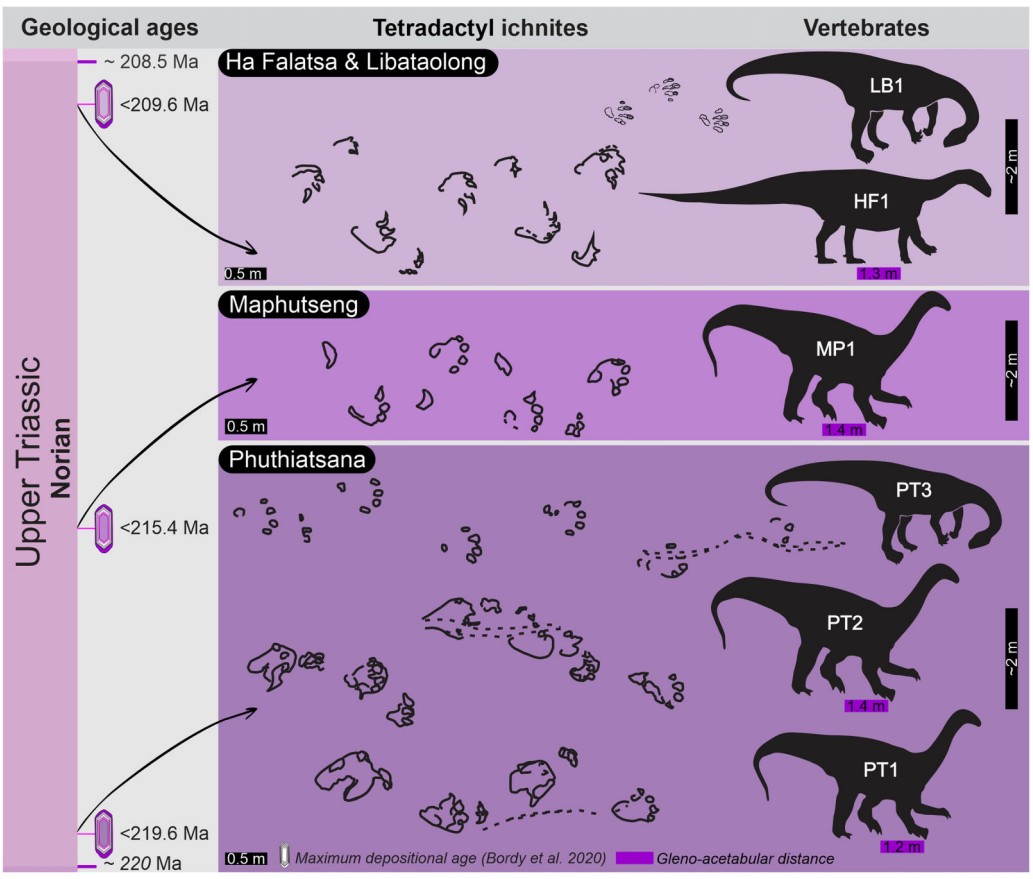

**Figure 16 Estimated prospective sauropodomorph trackmaker body size and stance based on the lower Elliot Formation tetradactyl trackways.** The hip heights and glenoacetabular distances are estimated and measured from trackway data (Table 1). Bipedal silhouette modified from *McPhee et al. (2017*, CC BY 4.0, Fig. 1; *Eucnemesaurus*). Quadrupedal silhouettes modified from *McPhee et al. (2017*; CC BY 4.0, Fig. 1; Blikanasaurus) and Wikipedia (*Melanorosaurus*). Scale for x and y axis the same.

medially, and this may be used to accommodate heavier body masses in the three central metatarsals (*Galton & Van Heerden, 1985*).

*Blikanasaurus* pedal morphology may fit the fleshier and robustly clawed and younger *Tetrasauropus* tracks, even though (i) no manus is known from *Blikanasaurus*; and (ii) *Tetrasauropus* manus tracks, particularly the manual claw traces, seem to fit the features of a melanosaurid trackmaker. Indeed, the pedal impressions of the *Tetrasauropus* indicate a possible pedal skeleton that shows a mosaic of sauropodomorph and sauropod traits (*i.e.*, robust first metatarsal and sickle-shaped and laterally compressed ungual of digits II and III; *Wilson & Sereno, 1998*; *Upchurch, Barrett & Galton, 2007*). Although, the sickle-shaped unguals in digits I–III have an orientation 180° opposite that of eusauropods (*Wilson, 2005*; *Upchurch, Barrett & Galton, 2007*). Particularly noteworthy is that *Blikanasaurus'* stout foot has a first pedal ungual that is longer than the first metatarsal (*McPhee et al., 2017*) and digits I and II are larger than III and IV, which would fit well with the *Tetrasauropus* digit traces. It should be noted that a yet-to-be named lEF basal

sauropodomorph (SAM-PK 382) is also characterized by a stout foot but with the pedal metatarsal I equal in size to ungual I (*McPhee et al., 2017*). *T. unguiferus* manus tracks, with strong digit I (*Wilson, 2005*; *D'Orazi Porchetti & Nicosia, 2007*), also echoes the robust digit I in basal sauropodomorphs, which is commonly considered non-weight bearing (*Upchurch, 1994*; *Galton & Upchurch, 2004*). The wide stance of *Tetrasauropus'* pes and manus tracks do indicate a flexed limb posture (*i.e.*, as in Lessemsaurids; *Ellenberger, 1970*; *Lockley & Meyer, 2000*; *Sander & Lallensack, 2018*). Altogether, the *Tetrasauropus* trackway illustrates a trackmaker with a mix of primitive and derived characters.

Lastly, the depth-pressure patterns noted in the trackways relative to the body fossil record provide food for thought. For instance, in *Tetrasauropus*, the depth of the impressions is usually greatest in the digit IV (the external margin of the pes), and as noted for more derived sauropods (*Carrano, 2005*), but the available and slender digit IV of the lEF pedal body fossils (Fig. 15), suggests this digit is, generally, less weight-bearing in comparison to the more robust central elements. The latter is reflected more strongly in the weight-bearing 'axis' around the central digits in *Pseudotetrasauropus*. Equally, the pedal record shows relatively robust digit I (Fig. 15) compared to the other digits, and yet our corresponding track record does attest to a disproportionate depth-pressure pattern to suggest this digit bore weight more substantially than the other digits in either *Tetrasauropus* or *Pseudotetrasauropus*. The latter shows a claw impression that may be lightly impressed (*e.g.*, *Pseudotetrasauropus* PT1). In fact, all tracks seem to suggest weight born more routinely through the central elements and the plausibility of a fleshy sole to distribute loading and locomotive pressures.

## CONCLUSIONS

This study on Late Triassic large tetradactyl footprint assemblages in southern Africa demonstrates, and more importantly, validates the original ideas of Paul Ellenberger, namely that: (a) tetradactyl tracks with 43–56 cm in length were commonly made in this region of southwestern Gondwana and, (b) their affinity to basal sauropodomorph reinforces the notion that the sauropods' ancestral forms had already evolved into large-bodied dinosaurs by the Norian. Moreover, the tracks also show that the Norian trackmakers used both bipedal and quadrupedal locomotion styles throughout a c. 10 Ma interval in the Late Triassic (Figs. 1, 15, 16). Speculation as to the adoption of quadrupedal and bipedal stances may be related to terrain and substrate conditions (*i.e.*, Phuthiatsana *Pseudotetrasauropus* isp. PT2 and PT3), and facultative quadrupedalism would have supported the ability to increase food acquisition without compromising biomechanics (as suggested by *McPhee et al., 2017*). The role of the substrate in the preservation of track morphology, behavior, and weight-distribution in the autopods has been thoroughly assessed. These exhibit contradictory evidence with the—albeit limited—pedal body fossil record, and we propose that the trackmakers likely had a fatty plantar pad that distributed pressures in the underfoot during locomotion. Finally, unique, and rare drag traces, of possible tail and manus digits, echo changes in locomotory habit and probably reflect the adaptation of the trackmaker to variations in substrate rheology and terrain topography. To-date, these are the oldest drag traces attributed to basal sauropodomorphs and assist in

validating their limb movements in the early Mesozoic, an important time in early dinosaur evolution.

## ABBREVIATIONS

| | |
|---|---|
| **RP** | Right pes |
| **LP** | Left pes |
| **RM** | Right manus |
| **LM** | Left manus |
| **DM** | Drag trace |
| **Gp** | Group |
| **Fm** | Formation |
| **Ma** | Million years |
| **c.** | *circa* |
| **PT** | Phuthiatsana |
| **MP** | Maphutseng |
| **HF** | Ha falatsa |
| **LB** | Libataolong |
| **Morija M&A** | Morija museum and archives |
| **UoM** | University of montpellier |

## ACKNOWLEDGEMENTS

This manuscript greatly benefitted from the advice, discussion, and comments of Blair McPhee (especially; so many thanks, Blataro!), Fabien Knoll, and Paul Olsen. This project would not have been as productive without the contributions of UCT Seds-Palaeo lab team members, particularly by Mhairi Reid (field work and examination of drag traces), Riyaad Mukaddam and Akhil Rampersadh (photographing cast materials). EB thanks the ongoing research support and permitting from Mme' Matsosane Molibeli (Lesotho Ministry of Tourism, Environment and Culture), Ntate Stephen Gill and Mme' Keletso Selialia Lesego (Lesotho Morija Museum and Archives); Ntate Lenoesa Oa Qeme (Moyeni Dinosaur Footprints visitors' centre); Ntate Kori Phakisi (Masitise Cave House) and Ntate David Ambrose (National University of Lesotho). Finally, warm appreciation to our thoughtful reviewers, Jens Lallensack and Lorenzo Marchetti, for their constructive comments on the manuscript.

### Funding

This project was made possible by the National Research Foundation and the DST-NRF Centre of Excellence in Palaeosciences (South Africa) grants of Emese M. Bordy. Postgraduate funding was provided by the DST-NRF Centre of Excellence in Palaeosciences and the Swiss National Science Foundation (SNF200021_192036) for

Miengah Abrahams and Lara Sciscio. The funders had no role in study design, data collection and analysis, decision to publish, or preparation of the manuscript.

### Grant Disclosures

The following grant information was disclosed by the authors:
National Research Foundation and the DST-NRF Centre of Excellence in Palaeosciences (South Africa).
DST-NRF Centre of Excellence in Palaeosciences.
Swiss National Science Foundation: SNF200021_192036.

### Competing Interests

The authors declare that they have no competing interests.

### Author Contributions

- Lara Sciscio conceived and designed the experiments, performed the experiments, analyzed the data, prepared figures and/or tables, authored or reviewed drafts of the article, and approved the final draft.
- Emese M. Bordy conceived and designed the experiments, performed the experiments, analyzed the data, prepared figures and/or tables, authored or reviewed drafts of the article, and approved the final draft.
- Martin G. Lockley conceived and designed the experiments, performed the experiments, analyzed the data, authored or reviewed drafts of the article, and approved the final draft.
- Miengah Abrahams conceived and designed the experiments, performed the experiments, analyzed the data, prepared figures and/or tables, authored or reviewed drafts of the article, and approved the final draft.

### Field Study Permissions

The following information was supplied relating to field study approvals (*i.e.*, approving body and any reference numbers):

Fieldwork was conducted under field permits (NR/M/E/10 and MTEC7/33) issued by the Lesotho Government Department of Mines and Geology and the Ministry of Tourism, Environment and Culture.

### Data Availability

Three-dimensional models are available at MorphoSource:
https://www.morphosource.org/projects/000526059
DOIs: 10.17602/M2/M526786, 10.17602/M2/M526780, 10.17602/M2/M526772, 10.17602/M2/M526758, 10.17602/M2/M526717, 10.17602/M2/M526681, 10.17602/M2/M526675, 10.17602/M2/M526657, 10.17602/M2/M526647, 10.17602/M2/M526635, 10.17602/M2/M526627, 10.17602/M2/M526621, 10.17602/M2/M526138, 10.17602/M2/M526118, 10.17602/M2/M526094.
## Supplemental Information

Supplemental information for this article can be found online at http://dx.doi.org/10.7717/peerj.15970#supplemental-information.

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
