# Peer review of "Basal sauropodomorph locomotion: ichnological lessons from the Late Triassic trackways of bipeds and quadrupeds (Elliot Formation, main Karoo Basin)"

_PeerJ, doi:10.7717/peerj.15970_

## Round 0.1 · original submission · Major Revisions

Two reviewers have provided nicely constructive critiques, and both offer strong enthusiasm for the paper. It clearly is a worthy contribution to the literature. They give some advice on amending the main text's wording, some changes to figures to make them clearer, and the need for explicit diagnoses + more robust ichnotaxonomy. These suggestions will strengthen the paper. Please implement them. I will check the revised MS (be sure to include a Track Changes version of MS with your submission) and it might not require re-review. Thank you!

·

Basic reporting

This is an important and long-awaited article describing material that helps to understand the locomotory evolution of sauropodomorphs. The figures are excellent. However, I found the manuscript a bit difficult to read in places, and I sometimes could not follow the line of argument and got confused about a number of things. See my line comments below for details:


l. 545: You are citing Rainforth (2003) for the statement that "digit IV is the longest" in Otozoum, but that paper (on p. 811) in fact states that digit III is longest?

l. 534: you are comparing the ichnogenera by giving measurements in cm. I worry that this can be meaningless when tracks differ in overall size, and suggest to use percentages instead.

l. 532: You state that these features are "not substrate related", but the presence of medial curvature in the claws could be an effect of the foot kinematics, and does not necessarily mean that claws are oriented like this in the animal. We see this in theropods quite often, too; and here we know that the unguals were straight in the skeletons. So I am not sure if straight claw marks (in contrast to medially deflected claw marks) is indeed an anatomically informative feature (if consistent across a larger sample, it might inform about differences in foot movements/locomotion).

What exactly do you mean with "digit length"? The length from the proximal phalangeal pad to the distal tip, the "free length", or the projections of the digits relative to each other?

l. 630 "a supinated hand" – that would mean the hand was facing backwards when quadrupedal. Do you mean "semi-supinated"?

l. 639 – species names have to be in italics

l. 645 "medial claw traces" – do you mean "medially deflected claw traces", or are the claws really restricted to the medial side of the track?

When you say "crouched", I am thinking of flexed limbs (as in a sitting position). Better use "sprawling" maybe?

l. 655 – Why is the forelimb rotation "likely restricted"? This is not explained. Also, it is not restricted at all in sauropods, where manus rotation is highly variable.

l. 656 – what does cushioning have to do with the external manus placement? Are you arguing that an external manus placement requires more cushioning, and if so, why is that?

l. 667 – why does it has to be flexed digits or knuckle walking? Is this really the most parsimonious explanation? Why can't we get something similar with a standard posture and foot movement? And I don't follow the "contra Thulborn, 1990" – what hypothesis of Thulborn (1990) are you referring to?

In some places in the manuscript, you use the word "modification" to describe track features. E.g, in l. 675, drag traces are "modifications" of the trackways. I am confused about this, because I don't see any original state that could possibly be "modified". Are you referring to some plesiomorphic condition here?

Or in l. 677: Why are substrate conditions "modifications" of the tracks? That does not make sense to me; the substrate conditions already existed before track formation. "Modification" can only occur after track formation, right?

l. 699: I do not understand how supination/pronation can produce an arcing trace. Why would rotation of the manus have an effect on the arcing of the swing? I do not understand the mechanism you have in mind here.

The 3D models are of very high quality. In some cases, e.g. Fig 11c or Fig 10c, there seem to be tiny floating parts included in the model that restrict the color scale so that the models appears green or blue only. To make use of the full color scale, you could use the "rescale to custom range" feature in Paraview. Furthermore, the color legends are often way too small to read, and seem to go from 0 to 1, which is not helpful (the actual depth in cm would be a more valuable information). We have instructions in our JPT paper for these things.

l. 730: What are "sediment disruption ridges"? I can't follow here.

I am also confused about the use of the terms "above" and "below". Does this mean "distal" and "proximal", perhaps?

l. 736: Again I am unsure what a "secondary sediment expulsion flap" exactly is, I never heard of this term. Is it a displacement rim? Since you are citing Gatesy (2003), do you mean "indirect" (i.e., Gatesy's term "indirect feature") instead of "secondary"? I don't understand in what sense the structure should be "secondary".

l. 737: Why does it highlight "its less weight-bearing nature"? For example, it could also have carried weight during mid-stance and then moved before lift-off to create the broad shape?

l. 741: You state "During movement, a medial arch provides both flexion
and rigidity to a foot (e.g., as in humans; Gwani et al., 2017)" – Here, and elsewhere in the manuscript, I am unsure what you mean with "arch". You seem to compare with the medial longitudinal arch in humans, but this arch is formed by the metatarsals and tarsals and therefore should not be visible in the tracks. Elsewhere you also mentioned "digit arc", but are you sure that there was a medial arc in the digits in the first place? Humans don't have one (I believe), and did anyone propose this for dinosaurs (if so, a citation would be great).


Jens N. Lallensack

Experimental design

see above

Validity of the findings

see above

·

Basic reporting

I commend the authors for the thorough study and description of this important footprint material attributed to sauropodomorphs, which has been long awaited, and I generally agree with their conclusions. However, I find that:

1) some background and data on sedimentology and depositional environment is lacking. So, I would suggest to add a brief section about the overall sedimentology of the lower Elliot Formation and some remarks on every site. This is relevant to understand the environment the animals walked in and the preservation of footprints;

2) the ichnotaxonomy section has an excessively informal structure and lacks diagnoses and comparisons with relevant material. So, I would suggest to add diagnoses for the two discussed ichnospecies Pseudotetrasauropus jaquesi and Tetrasauropus unguiferus, list the material for both and clearly differentiate these ichnospecies from each other and from other early sauropodomorph ichnotaxa including not only Otozoum but also Evazoum, Kalosauropus and Eosauropus. Moreover, the discussion would benefit from a comparison of Pseudotetrasauropus jaquesi with Brachychirotherium parvum (see Klein et al. 2006). In this sense, it would be interesting a discussion on the potential occurrence of a digit V basal pad in Pseudotetrasauropus. Moreover, a short discussion on whether to add both Pseudotetrasauropus and Tetrasauropus to the Otozoidae ichnofamily would be needed;

3) the drawings on orthophotos are not very visible. I suggest to show the outline only, the orthophoto is already shown aside so it is sufficient to understand the drawing.

Experimental design

The investigation is thorough and comprehensive. Nevertheless, is unclear:

1) how the measurements were taken (on the original specimens, on transparent film drawings or digitally and in this case, on what - scaled photos, ortophotos, drawings, 3D models... and also using which software);

2) whether there was a material selection by means of preservation and if only this material has been measured (as regards pes and manus parameters);

3) whether the measurements of the table are averages and in the case they are not, the complete measurements should be added to the supplementary material;

4) how the footprint preservation has been evaluated. In this sense, the use of the preservation scale is recommended, at least for the sections called "substrate consistency and impression depth" which really are about preservation.

Validity of the findings

no comment

---

## Round 0.2 · accepted · Accept

I have checked the rebuttal and tracked changes manuscript and am pleased by the attentive, reasonable revisions. I see nothing else that must be changed, or good reason for re-review. Please double-check that your references' DOIs are all completed; I see one that is not. Congratulations!! Thanks for this useful and interesting contribution.